# BECLR: Batch Enhanced Contrastive Few-Shot Learning

**Stylianos Poulakakis-Daktylidis**[1] **& Hadi Jamali-Rad**[1,2]
[1]Delft University of Technology (TU Delft), The Netherlands
[2]Shell Global Solutions International B.V., Amsterdam, The Netherlands
`{s.p.mrpoulakakis-daktylidis, h.jamalirad}@tudelft.nl`

## Abstract

Learning quickly from very few labeled samples is a fundamental attribute that separates machines and humans in the era of deep representation learning. Unsupervised few-shot learning (U-FSL) aspires to bridge this gap by discarding the reliance on annotations at training time. Intrigued by the success of contrastive learning approaches in the realm of U-FSL, we structurally approach their shortcomings in both pretraining and downstream inference stages. We propose a novel `Dynamic Clustered mEmory` (`DyCE`) module to promote a highly separable latent representation space for *enhancing positive sampling* at the pretraining phase and infusing implicit class-level insights into unsupervised contrastive learning. We then tackle the, somehow overlooked yet critical, issue of *sample bias* at the few-shot inference stage. We propose an iterative `Optimal Transport-based distribution Alignment` (`OpTA`) strategy and demonstrate that it efficiently addresses the problem, especially in low-shot scenarios where FSL approaches suffer the most from sample bias. We later on discuss that `DyCE` and `OpTA` are two intertwined pieces of a novel end-to-end approach (we coin as `BECLR`), constructively magnifying each other's impact. We then present a suite of extensive quantitative and qualitative experimentation to corroborate that `BECLR` sets a new state-of-the-art across ALL existing U-FSL benchmarks (to the best of our knowledge), and significantly outperforms the best of the current baselines (codebase available at GitHub).

## 1 Introduction

Achieving acceptable performance in deep representation learning comes at the cost of humongous data collection, laborious annotation, and excessive supervision. As we move towards more complex downstream tasks, this becomes increasingly prohibitive; in other words, supervised representation learning simply does not scale. In stark contrast, humans can quickly learn new tasks from a handful of samples, without extensive supervision. Few-shot learning (FSL) aspires to bridge this fundamental gap between humans and machines, using a suite of approaches such as metric learning (Wang et al., 2019; Bateni et al., 2020; Yang et al., 2020), meta-learning (Finn et al., 2017; Rajeswaran et al., 2019; Rusu et al., 2018), and probabilistic learning (Iakovleva et al., 2020; Hu et al., 2019; Zhang et al., 2021). FSL has shown promis-

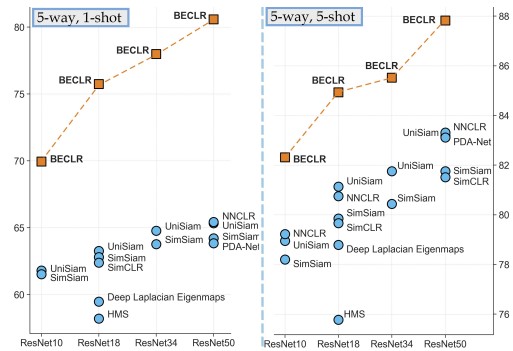

Figure 1: miniImageNet (5-way, 1-shot, left) and (5-way, 5-shot, right) accuracy in the U-FSL landscape. `BECLR` sets a new state-of-the-art in all settings by a significant margin.

ing results in a supervised setting so far on a number of benchmarks (Hu et al., 2022; Singh & Jamali-Rad, 2022; Hu et al., 2023b); however, the need for supervision still lingers on. This has led to the emergence of a new divide called unsupervised FSL (U-FSL). The stages of U-FSL are the same as their supervised counterparts: pretraining on a large dataset of *base* classes followed by fast adaptation and inference to unseen few-shot tasks (of *novel* classes). The extra challenge here is the absence of labels during pretraining. U-FSL approaches have gained an upsurge of attention most recently owing to their practical significance and close ties to self-supervised learning.

The goal of pretraining in U-FSL is to learn a feature extractor (a.k.a encoder) to capture the global structure of the unlabeled data. This is followed by fast adaptation of the frozen encoder to unseen tasks typically through a simple linear classifier (e.g., a logistic regression classifier). The body of literature here can be summarized into two main categories: (i) meta-learning-based pretraining where synthetic few-shot tasks resembling the downstream inference are generated for episodic training of the encoder (Hsu et al., 2018; Khodadadeh et al., 2019; 2020); (ii) (non-episodic) transfer-learning approaches where pretraining boils down to learning optimal representations suitable for downstream few-shot tasks (Medina et al., 2020; Chen et al., 2021b; 2022; Jang et al., 2022). Recent studies demonstrate that (more) complex meta-learning approaches are data-inefficient (Dhillon et al., 2019; Tian et al., 2020), and that the transfer-learning-based methods outperform their meta-learning counterparts. More specifically, the state-of-the-art in this space is currently occupied by approaches based on *contrastive learning*, from the transfer-learning divide, achieving top performance across a wide variety of benchmarks (Chen et al., 2021a; Lu et al., 2022). The underlying idea of contrastive representation learning (Chen et al., 2020a; He et al., 2020) is to attract "positive" samples in the representation space while repelling "negative" ones. To efficiently materialize this, some contrastive learning approaches incorporate memory queues to alleviate the need for larger batch sizes (Zhuang et al., 2019; He et al., 2020; Dwibedi et al., 2021; Jang et al., 2022).

**Key Idea I:** *Going beyond instance-level contrastive learning.* Operating under the unsupervised setting, contrastive FSL approaches typically enforce consistency only at the *instance level*, where each image within the batch and its augmentations correspond to a unique class, which is an unrealistic but seemingly unavoidable assumption. The pitfall here is that potential positive samples present within the same batch might then be repelled in the representation space, hampering the overall performance. We argue that infusing a semblance of class (or membership)-level insights into the unsupervised contrastive paradigm is essential. Our key idea to address this is extending the concept of memory queues by introducing inherent membership clusters represented by dynamically updated prototypes, while circumventing the need for large batch sizes. This enables the proposed pretraining approach to sample more meaningful positive pairs owing to a novel `Dynamic Clustered mEmory (DyCE)` module. While maintaining a fixed memory size (same as queues), `DyCE` efficiently constructs and dynamically updates separable memory clusters.

**Key Idea II:** *Addressing inherent sample bias in FSL.* The base (pretraining) and novel (inference) classes are either mutually exclusive classes of the same dataset or originate from different datasets - both scenarios are investigated in this paper (in Section 5). This distribution shift poses a significant challenge at inference time for the swift adaptation to the novel classes. This is further aggravated due to access to only a few labeled (a.k.a *support*) samples within the few-shot task because the support samples are typically not representative of the larger unlabeled (a.k.a *query*) set. We refer to this phenomenon as *sample bias*, highlighting that it is overlooked by most (U-)FSL baselines. To address this issue, we introduce an `Optimal Transport-based distribution Alignment (OpTA)` add-on module within the supervised inference step. `OpTA` imposes no additional learnable parameters, yet efficiently aligns the representations of the labeled support and the unlabeled query sets, right before the final supervised inference step. Later on in Section 5, we demonstrate that these two novel modules (`DyCE` and `OpTA`) are actually intertwined and amplify one another. Combining these two key ideas, we propose an end-to-end U-FSL approach coined as `Batch-Enhanced Contrastive LeaRning (BECLR)`. Our main contributions can be summarized as follows:

I. We introduce `BECLR` to structurally address two key shortcomings of the prior art in U-FSL at pretraining and inference stages. At pretraining, we propose to infuse implicit class-level insights into the contrastive learning framework through a novel dynamic clustered memory (coined as `DyCE`). Iterative updates through `DyCE` help establish a highly separable partitioned latent space, which in turn promotes more meaningful positive sampling.

II. We then articulate and address the inherent sample bias in (U-)FSL through a novel add-on module (coined as `OpTA`) at the inference stage of `BECLR`. We show that this strategy helps mitigate the distribution shift between query and support sets. This manifests its significant impact in low-shot scenarios where FSL approaches suffer the most from sample bias.

III. We perform *extensive* experimentation to demonstrate that `BECLR` sets a new state-of-the-art in ALL established U-FSL benchmarks; e.g. miniImageNet (see Fig. 1), tieredImageNet, CIFAR-FS, FC100, outperforming ALL existing baselines by a significant margin (up to 14%, 12%, 15%, 5.5% in 1-shot settings, respectively), to the best of our knowledge.

## 2 RELATED WORK

**Self-Supervised Learning (SSL).** It has been approached from a variety of perspectives (Balestriero et al., 2023). Deep metric learning methods (Chen et al., 2020a; He et al., 2020; Caron et al., 2020; Dwibedi et al., 2021), build on the principle of a contrastive loss and encourage similarity between semantically transformed views of an image. Redundancy reduction approaches (Zbontar et al., 2021; Bardes et al., 2021) infer the relationship between inputs by analyzing their cross-covariance matrices. Self-distillation methods (Grill et al., 2020; Chen & He, 2021; Oquab et al., 2023) pass different views to two separate encoders and map one to the other via a predictor. Most of these approaches construct a contrastive setting, where a symmetric or (more commonly) asymmetric Siamese network (Koch et al., 2015) is trained with a variant of the infoNCE(Oord et al., 2018) loss.

**Unsupervised Few-Shot Learning (U-FSL).** The objective here is to pretrain a model from a large *unlabeled* dataset of base classes, akin to SSL, but tailored so that it can quickly generalize to unseen downstream FSL tasks. Meta-learning approaches (Lee et al., 2020; Ye et al., 2022) generate synthetic learning episodes for pretraining, which mimic downstream FSL tasks. Here, PsCo (Jang et al., 2022) utilizes a student-teacher momentum network and optimal transport for creating pseudo-supervised episodes from a memory queue. Despite its elegant form, meta-learning has been shown to be data-inefficient in U-FSL (Dhillon et al., 2019; Tian et al., 2020). On the other hand, transfer-learning approaches (Li et al., 2022; Antoniou & Storkey, 2019; Wang et al., 2022a), follow a simpler non-episodic pretraining, focused on representation quality. Notably, contrastive learning methods, such as PDA-Net (Chen et al., 2021a) and UniSiam (Lu et al., 2022), currently hold the state-of-the-art. Our proposed approach also operates within the contrastive learning premise, but also employs a dynamic clustered memory module (`DyCE`) for infusing membership/class-level insights within the instance-level contrastive framework. Here, SAMPTransfer (Shirekar et al., 2023) takes a different path and tries to ingrain *implicit* global membership-level insights through message passing on a graph neural network; however, the computational burden of this approach significantly hampers its performance with (and scale-up to) deeper backbones than `Conv4`.

**Sample Bias in (U-)FSL.** Part of the challenge in (U-)FSL lies in the domain difference between base and novel classes. To make matters worse, estimating class distributions only from a few support samples is inherently biased, which we refer to as *sample bias*. To address sample bias, Chen et al. (2021a) propose to enhance the support set with additional base-class images, Xu et al. (2022) project support samples farther from the task centroid, while Yang et al. (2021) use a calibrated distribution for drawing more support samples, yet all these methods are dependent on base-class characteristics. On the other hand, Ghaffari et al. (2021); Wang et al. (2022b) utilize Firth bias reduction to alleviate the bias in the logistic classifier itself, yet are prone to over-fitting. In contrast, the proposed `OpTA` module requires no fine-tuning and does not depend on the pretraining dataset.

## 3 PROBLEM STATEMENT: UNSUPERVISED FEW-SHOT LEARNING

We follow the most commonly adopted setting in the literature (Chen et al., 2021a;b; Lu et al., 2022; Jang et al., 2022), which consists of: an unsupervised pretraining, followed by a supervised inference (a.k.a fine-tuning) strategy. Formally, we consider a large unlabeled dataset $\mathcal{D}_{tr} = \{\boldsymbol{x}_i\}$ of so-called "base" classes for pretraining the model. The inference phase then involves transferring the model to unseen few-shot downstream tasks $\mathcal{T}_i$, drawn from a smaller labeled test dataset of so-called "novel" classes $\mathcal{D}_{tst} = \{(\boldsymbol{x}_i, y_i)\}$, with $y_i$ denoting the label of sample $\boldsymbol{x}_i$. Each task $\mathcal{T}_i$ is composed of two parts $[\mathcal{S}, \mathcal{Q}]$: (i) the support set $\mathcal{S}$, from which the model learns to adapt to the novel classes, and (ii) the query set $\mathcal{Q}$, on which the model is evaluated. The support set $\mathcal{S} = \{\boldsymbol{x}_i^s, y_i^s\}_{i=1}^{NK}$ is constructed by drawing $K$ labeled random samples from $N$ different classes, resulting in the so-called ($N$-way, $K$-shot) setting. The query set $\mathcal{Q} = \{\boldsymbol{x}_j^q\}_{j=1}^{NQ}$ contains $NQ$ (with $Q > K$) unlabeled samples. The base and novel classes are mutually exclusive, i.e., the distributions of $\mathcal{D}_{tr}$ and $\mathcal{D}_{tst}$ are different.

## 4 PROPOSED METHOD: BECLR

### 4.1 UNSUPERVISED PRETRAINING

We build the unsupervised pretraining strategy of `BECLR` following contrastive representation learning. The core idea here is to efficiently attract "positive" samples (i.e., augmentations of the same image) in the representation space, while repelling "negative" samples. However, traditional contrastive learning approaches address this at the *instance level*, where each image within the batch

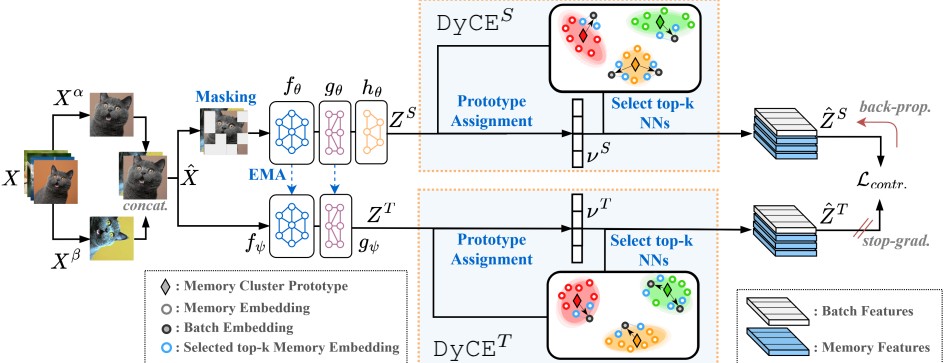

Figure 2: Overview of the proposed pretraining framework of BECLR. Two augmented views of the batch images $\boldsymbol{X}^{\{\alpha,\beta\}}$ are both passed through a student-teacher network followed by the DyCE memory module. DyCE enhances the original batch with meaningful positives and dynamically updates the memory partitions.

has to correspond to a unique class (a statistically unrealistic assumption!). As a result, potential positives present within a batch might be treated as negatives, which can have a detrimental impact on performance. A common strategy to combat this pitfall (also to avoid prohibitively large batch sizes) is to use a memory queue (Wu et al., 2018; Zhuang et al., 2019; He et al., 2020). Exceptionally, PsCo (Jang et al., 2022) uses optimal transport to sample from a first-in-first-out memory queue and generate pseudo-supervised few-shot tasks in a meta-learning-based framework, whereas NNCLR (Dwibedi et al., 2021) uses a nearest-neighbor approach to generate more meaningful positive pairs in the contrastive loss. However, these memory queues are still oblivious to global memberships (i.e., class-level information) in the latent space. Instead, we propose to infuse membership/class-level insights through a novel memory module (DyCE) within the pretraining phase of the proposed end-to-end approach: BECLR. Fig. 2 provides a schematic illustration of the proposed contrastive pretraining framework within BECLR, and Fig. 3 depicts the mechanics of DyCE.

**Pretraining Strategy of BECLR.** The pretraining pipeline of BECLR is summarized in Algorithm 1, and a Pytorch-like pseudo-code can be found in Appendix E. Let us now walk you through the algorithm. Let $\zeta^a, \zeta^b \sim \mathcal{A}$ be two randomly sampled data augmentations from the set of all available augmentations, $\mathcal{A}$. The mini-batch can then be denoted as $\hat{\boldsymbol{X}} = [\hat{\boldsymbol{x}}_i]_{i=1}^{2B} = [[\zeta^a(\boldsymbol{x}_i)]_{i=1}^B, [\zeta^b(\boldsymbol{x}_i)]_{i=1}^B]$, where $B$ the origi-

---

**Algorithm 1:** Pretraining of BECLR

**Require:** $\mathcal{A}, \theta, \psi, f_\theta, f_\psi, g_\theta, g_\psi, h_\theta, \mu, m, \text{DyCE}(\cdot)$
1   $\hat{\boldsymbol{X}} = [\zeta^\alpha(\boldsymbol{X}), \zeta^\beta(\boldsymbol{X})]$ for $\zeta^\alpha, \zeta^\beta \sim \mathcal{A}$
2   $\boldsymbol{Z}^S = h_\theta \circ g_\theta \circ f_\theta(\mu(\hat{\boldsymbol{X}}))$
3   $\boldsymbol{Z}^T = g_\psi \circ f_\psi(\hat{\boldsymbol{X}})$
4   $\hat{\boldsymbol{Z}}^S, \hat{\boldsymbol{Z}}^T = \text{DyCE}^S(\boldsymbol{Z}^S), \text{DyCE}^T(\boldsymbol{Z}^T)$
5   Compute loss: $\mathcal{L}_{contr.}$ using Eq. 3 on $\hat{\boldsymbol{Z}}^S, \hat{\boldsymbol{Z}}^T$
6   Update: $\theta \leftarrow \theta - \nabla\mathcal{L}_{contr.}, \; \psi \leftarrow m\psi + (1-m)\theta$

---

nal batch size (line 1, Algorithm 1). As shown in Fig. 2, we adopt a student-teacher (a.k.a Siamese) asymmetric momentum architecture similar to Grill et al. (2020); Chen & He (2021). Let $\mu(\cdot)$ be a patch-wise masking operator, $f(\cdot)$ the backbone feature extractor (ResNets (He et al., 2016) in our case), and $g(\cdot), h(\cdot)$ projection and prediction multi-layer perceptrons (MLPs), respectively. The teacher weight parameters ($\psi$) are an exponential moving averaged (*EMA*) version of the student parameters ($\theta$), i.e., $\psi \leftarrow m\psi + (1-m)\theta$, as in Grill et al. (2020), where $m$ the momentum hyper-parameter, while $\theta$ are updated through stochastic gradient descent (SGD). The student and teacher representations $\boldsymbol{Z}^S$ and $\boldsymbol{Z}^T$ (both of size $2B \times d$, with $d$ the latent embedding dimension) can then be obtained as follows: $\boldsymbol{Z}^S = h_\theta \circ g_\theta \circ f_\theta(\mu(\hat{\boldsymbol{X}}))$, $\boldsymbol{Z}^T = g_\psi \circ f_\psi(\hat{\boldsymbol{X}})$ (lines 2,3).

Upon extracting $\boldsymbol{Z}^S$ and $\boldsymbol{Z}^T$, they are fed into the proposed dynamic memory module (DyCE), where enhanced versions of the batch representations $\hat{\boldsymbol{Z}}^S, \hat{\boldsymbol{Z}}^T$ (both of size $2B(k+1) \times d$, with $k$ denoting the number of selected nearest neighbors) are generated (line 4). Finally, we apply the contrastive loss in Eq. 3 on the enhanced batch representations $\hat{\boldsymbol{Z}}^S, \hat{\boldsymbol{Z}}^T$ (line 5). Upon finishing unsupervised pretraining, only the student encoder ($f_\theta$) is kept for the subsequent inference stage.

**Dynamic Clustered Memory (DyCE).** *How do we manage to enhance the batch with meaningful true positives in the absence of labels?* We introduce DyCE: a novel dynamically updated clustered memory to moderate the representation space during training, while infusing a semblance of class-cognizance. We demonstrate later on in Section 5 that the design choices in DyCE have a significant impact on both pretraining performance as well as the downstream few-shot classification.

Let us first establish some notation. We consider a memory unit $\mathcal{M}$ capable of storing up to $M$ latent embeddings (each of size $d$). To accommodate clustered memberships within DyCE, we consider up to $P$ partitions (or clusters) $\mathcal{P}_i$ in $\mathcal{M} = [\mathcal{P}_1, \ldots, \mathcal{P}_P]$, each of which is represented by a prototype stored in $\mathbf{\Gamma} = [\boldsymbol{\gamma}_1, \ldots, \boldsymbol{\gamma}_P]$. Each prototype $\boldsymbol{\gamma}_i$ (of size $1 \times d$) is the average of the latent embeddings stored in partition $\mathcal{P}_i$. As shown in Fig. 3 and in Algorithm 2, DyCE consists of two informational paths: (i) the top-$k$ neighbor selection and batch enhancement path (bottom branch of the figure), which uses the current state of $\mathcal{M}$ and $\mathbf{\Gamma}$; (ii) the iterative memory updating via dynamic clustering path (top branch). DyCE takes as input student or teacher embeddings (we use $\boldsymbol{Z}$ here, for brevity) and returns the enhanced versions $\hat{\boldsymbol{Z}}$. We also allow for an adaptation period epoch $<$ epoch$_{\text{thr}}$ (empirically 20-50 epochs), during which path (i) is not activated and the training batch is not enhanced. To further elaborate, path (i) starts with assigning each $\boldsymbol{z}_i \in \boldsymbol{Z}$ to its nearest (out of $P$) memory prototype $\boldsymbol{\gamma}_{\nu_i}$ based on the Euclidean distance $\langle \cdot \rangle$. $\boldsymbol{\nu}$ is a vector of indices (of size $2B \times 1$) that contains these prototype assignments for all batch embeddings (line 4, Algorithm 2). Next (in line 5), we se-

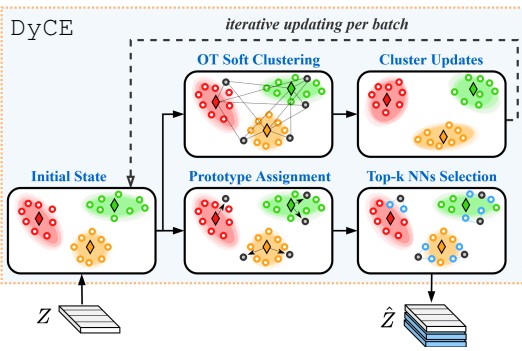

Figure 3: Overview of the proposed dynamic clustered memory (DyCE) and its two informational paths.

---

**Algorithm 2: DyCE**

**Require:** epoch$_{\text{thr}}$, $\mathcal{M}$, $\mathbf{\Gamma}$, $\boldsymbol{Z}$, $B$, $k$
1 **if** $|\mathcal{M}| = M$ **then**
2    // Path (i): top-$k$ and batch enhancement)
3    **if** epoch $\geq$ epoch$_{\text{thr}}$ **then**
4      $\boldsymbol{\nu} = [\nu_i]_{i=1}^{2B} = \left[\operatorname*{argmin}_{j \in [P]} \langle \boldsymbol{z}_i, \boldsymbol{\gamma}_j \rangle \right]_{i=1}^{2B}$
5      $\boldsymbol{Y}_i \leftarrow$ top-k$(\{\boldsymbol{z}_i, \mathcal{P}_{\nu_i}\}), \forall i \in [2B]$
6      $\hat{\boldsymbol{Z}} = [\boldsymbol{Z}, \boldsymbol{Y}_1, \ldots, \boldsymbol{Y}_{2B}]$
7    // Path (ii): iterative memory updating
8    Find optimal plan between $\boldsymbol{Z}, \mathbf{\Gamma}$: $\boldsymbol{\pi}^\star \leftarrow$ Solve Eq. 2
9    Update $\mathcal{M}$ with new $\boldsymbol{Z}$: $\mathcal{M} \leftarrow$ update$(\mathcal{M}, \boldsymbol{\pi}^\star, \boldsymbol{Z})$
10    Discard $2B$ oldest batch embeddings: dequeue$(\mathcal{M})$
11 **else**
12    Store new batch: $\mathcal{M} \leftarrow$ store$(\mathcal{M}, \boldsymbol{Z})$
**Return:** $\hat{\boldsymbol{Z}}$

---

lect the $k$ most similar memory embeddings to $\boldsymbol{z}_i$ from the memory partition corresponding to its assigned prototype ($\mathcal{P}_{\nu_i}$) and store them in $\boldsymbol{Y}_i$ (of size $k \times d$). Finally (in line 6), all $\boldsymbol{Y}_i, \forall i \in [2B]$ are concatenated into the enhanced batch $\hat{\boldsymbol{Z}} = [\boldsymbol{Z}, \boldsymbol{Y}_1, \ldots, \boldsymbol{Y}_{2B}]$ of size $L \times d$ (where $L = 2B(k+1)$). Path (ii) addresses the iterative memory updating by casting it into an optimal transport problem (Cuturi, 2013) given by:

$$\mathbf{\Pi}(\boldsymbol{r}, \boldsymbol{c}) = \left\{ \boldsymbol{\pi} \in \mathbb{R}_+^{2B \times P} \mid \boldsymbol{\pi} \mathbf{1}_P = \boldsymbol{r}, \ \boldsymbol{\pi}^\top \mathbf{1}_{2B} = \boldsymbol{c}, \ \boldsymbol{r} = \mathbf{1} \cdot {}^1/_{2B}, \ \boldsymbol{c} = \mathbf{1} \cdot {}^1/_P \right\}, \tag{1}$$

to find a transport plan $\boldsymbol{\pi}$ (out of $\mathbf{\Pi}$) mapping $\boldsymbol{Z}$ to $\mathbf{\Gamma}$. Here, $\boldsymbol{r} \in \mathbb{R}^{2B}$ denotes the distribution of batch embeddings $[\boldsymbol{z}_i]_{i=1}^{2B}$, $\boldsymbol{c} \in \mathbb{R}^P$ is the distribution of memory cluster prototypes $[\boldsymbol{\gamma}_i]_{i=1}^P$. The last two conditions in Eq. 1 enforce equipartitioning (i.e., uniform assignment) of $\boldsymbol{Z}$ into the $P$ memory partitions/clusters. Obtaining the optimal transport plan, $\boldsymbol{\pi}^\star$, can then be formulated as:

$$\boldsymbol{\pi}^\star = \operatorname*{argmin}_{\boldsymbol{\pi} \in \mathbf{\Pi}(\boldsymbol{r}, \boldsymbol{c})} \langle \boldsymbol{\pi}, \boldsymbol{D} \rangle_F - \varepsilon \mathbb{H}(\boldsymbol{\pi}), \tag{2}$$

and solved using the Sinkhorn-Knopp (Cuturi, 2013) algorithm (line 8). Here, $\boldsymbol{D}$ is a pairwise distance matrix between the elements of $\boldsymbol{Z}$ and $\mathbf{\Gamma}$ (of size $2B \times P$), $\langle \cdot \rangle_F$ denotes the Frobenius dot product, $\varepsilon$ is a regularisation term, and $\mathbb{H}(\cdot)$ is the Shannon entropy. Next (in line 9), we add the embeddings of the current batch $\boldsymbol{Z}$ to $\mathcal{M}$ and use $\boldsymbol{\pi}^\star$ for updating the partitions $\mathcal{P}_i$ and prototypes $\mathbf{\Gamma}$ (using EMA for updating). Finally, we discard the $2B$ oldest memory embeddings (line 10).

**Loss Function.** The popular infoNCE loss (Oord et al., 2018) is the basis of our loss function, yet recent studies (Poole et al., 2019; Song & Ermon, 2019) have shown that it is prone to high bias, when the batch size is small. To address this, we adopt a variant of infoNCE, which maximizes the same mutual information objective, but has been shown to be less biased (Lu et al., 2022):

$$\mathcal{L}_{contr.} = \frac{1}{L} \sum_{i=1}^{L/2} \left( \mathrm{d}[\boldsymbol{z}_i^S, \boldsymbol{z}_i^{T+}] + \mathrm{d}[\boldsymbol{z}_i^{S+}, \boldsymbol{z}_i^T] \right) - \lambda \log \left( \frac{1}{L} \sum_{i=1}^{L} \sum_{j \neq i, i^+} \exp(\mathrm{d}[\boldsymbol{z}_i^S, \boldsymbol{z}_j^S]/\tau) \right), \tag{3}$$

where $\tau$ is a temperature parameter, $\mathrm{d}[\cdot]$ is the negative cosine similarity, $\lambda$ is a weighting hyperparameter, $L$ is the enhanced batch size and $\boldsymbol{z}_i^+$ stands for the latent embedding of the positive sample,

Figure 4: Overview of the inference strategy of BECLR. Given a test episode, the support ($\mathcal{S}$) and query ($\mathcal{Q}$) sets are passed to the pretrained feature extractor ($f_\theta$). OpTA aligns support prototypes and query features.

corresponding to sample $i$. Following (Chen & He, 2021), to further boost training performance, we pass both views through both the student and the teacher. The first term in Eq. 3 operates on positive pairs, and the second term pushes each representation away from all other batch representations.

## 4.2 SUPERVISED INFERENCE

Supervised inference (a.k.a fine-tuning) usually combats the distribution shift between training and test datasets. However, the limited number of support samples (in FSL tasks) at test time leads to a significant performance degradation due to the so-called *sample bias* (Cui & Guo, 2021; Xu et al., 2022). This issue is mostly disregarded in recent state-of-the-art U-FSL baselines (Chen et al., 2021a; Lu et al., 2022; Hu et al., 2023a). As part of the inference phase of BECLR, we propose a simple, yet efficient, add-on module (coined as OpTA) for aligning the distributions of the query ($\mathcal{Q}$) and support ($\mathcal{S}$) sets, to structurally address sample bias. Notice that OpTA is not a learnable module and that there are no model updates nor any fine-tuning involved in the inference stage of BECLR.

**Optimal Transport-based Distribution Alignment (OpTA).** Let $\mathcal{T} = \mathcal{S} \cup \mathcal{Q}$ be a downstream few-shot task. We first extract the support $\boldsymbol{Z}^{\mathcal{S}} = f_\theta(\mathcal{S})$ (of size $NK \times d$) and query $\boldsymbol{Z}^{\mathcal{Q}} = f_\theta(\mathcal{Q})$ (of size $NQ \times d$) embeddings and calculate the support set prototypes $\boldsymbol{P}^{\mathcal{S}}$ (class averages of size $N \times d$). Next, we find the optimal transport plan ($\boldsymbol{\pi}^\star$) between $\boldsymbol{P}^{\mathcal{S}}$ and $\boldsymbol{Z}^{\mathcal{Q}}$ using Eq. 2, with $\boldsymbol{r} \in \mathbb{R}^{NQ}$ the distribution of $\boldsymbol{Z}^{\mathcal{Q}}$ and $\boldsymbol{c} \in \mathbb{R}^N$ the distribution of $\boldsymbol{P}^{\mathcal{S}}$. Finally, we use $\boldsymbol{\pi}^\star$ to map the support set prototypes onto the region occupied by the query embeddings:

$$\hat{\boldsymbol{P}}^{\mathcal{S}} = \hat{\boldsymbol{\pi}}^{\star T} \boldsymbol{Z}^{\mathcal{Q}}, \quad \hat{\boldsymbol{\pi}}_{i,j}^\star = \frac{\boldsymbol{\pi}_{i,j}^\star}{\sum_j \boldsymbol{\pi}_{i,j}^\star}, \forall i \in [NQ], j \in [N], \tag{4}$$

where $\hat{\boldsymbol{\pi}}^\star$ is the normalized transport plan and $\hat{\boldsymbol{P}}^{\mathcal{S}}$ are the transported support prototypes. Finally, we fit a logistic regression classifier on $\hat{\boldsymbol{P}}^{\mathcal{S}}$ to infer on the unlabeled query set. We show in Section 5 that OpTA successfully minimizes the distribution shift (between support and query sets) and contributes to the overall significant performance margin BECLR offers against the best existing baselines. Note that we iteratively perform $\delta$ consecutive passes of OpTA, where $\hat{\boldsymbol{P}}^{\mathcal{S}}$ acts as the input of the next pass. Notably, OpTA can straightforwardly be applied on top of any U-FSL approach. An overview of OpTA and the proposed inference strategy is illustrated in Fig. 4.

**Remark:** OpTA operates on two distributions and relies on the total number of unlabeled query samples being larger than the total number of labeled support samples ($|\mathcal{Q}| > |\mathcal{S}|$) for reasonable distribution mapping, which is also the standard convention in the U-FSL literature. That said, OpTA would still perform on imbalanced FSL tasks as long as the aforementioned condition is met.

## 5 EXPERIMENTS

In this section, we rigorously study the performance of the proposed approach both quantitatively as well as qualitatively by addressing the following three main questions:
[**Q1**] How does BECLR perform against the state-of-the-art for *in-domain* and *cross-domain* settings?
[**Q2**] Does DyCE affect pretraining performance by establishing *separable memory partitions*?
[**Q3**] Does OpTA address the *sample bias* via the proposed distribution alignment strategy?

We use PyTorch (Paszke et al., 2019) for all implementations. Elaborate implementation and training details are discussed in the supplementary material, in Appendix A.

**Benchmark Datasets.** We evaluate BECLR in terms of its in-domain performance on the two most widely adopted few-shot image classification datasets: miniImageNet (Vinyals et al., 2016) and tieredImageNet (Ren et al., 2018). Additionally, for the in-domain setting, we also evaluate on two

Table 1: Accuracies (in % ± std) on miniImageNet and tieredImageNet compared against unsupervised (Unsup.) and supervised (Sup.) baselines. Backbones: RN: Residual network.[†]: denotes our reproduction. [*]: denotes extra synthetic training data used. Style: **best** and second best.

| Method | Backbone | Setting | miniImageNet | | tieredImageNet | |
|---|---|---|---|---|---|---|
| | | | 5-way 1-shot | 5-way 5-shot | 5-way 1-shot | 5-way 5-shot |
| SwAV[†] (Caron et al., 2020) | RN18 | Unsup. | 59.84 ±0.52 | 78.23 ±0.26 | 65.26 ±0.53 | 81.73 ±0.24 |
| NNCLR[†] (Dwibedi et al., 2021) | RN18 | Unsup. | 63.33 ±0.53 | 80.75 ±0.25 | 65.46 ±0.55 | 81.40 ±0.27 |
| CPNWCP (Wang et al., 2022a) | RN18 | Unsup. | 53.14 ±0.62 | 67.36 ±0.5 | 45.00 ±0.19 | 62.96 ±0.19 |
| HMS (Ye et al., 2022) | RN18 | Unsup. | 58.20 ±0.23 | 75.77 ±0.16 | 58.42 ±0.25 | 75.85 ±0.18 |
| SAMPTransfer[†] (Shirekar et al., 2023) | RN18 | Unsup. | 45.75 ±0.77 | 68.33 ±0.66 | 42.32 ±0.75 | 53.45 ±0.76 |
| PsCo[†] (Jang et al., 2022) | RN18 | Unsup. | 47.24 ±0.76 | 65.48 ±0.68 | 54.33 ±0.54 | 69.73 ±0.49 |
| UniSiam + dist (Lu et al., 2022) | RN18 | Unsup. | 64.10 ±0.36 | 82.26 ±0.25 | 67.01 ±0.39 | 84.47 ±0.28 |
| Meta-DM + UniSiam + dist[*] (Hu et al., 2023a) | RN18 | Unsup. | 65.64 ±0.36 | 83.97 ±0.25 | 67.11 ±0.40 | 84.39 ±0.28 |
| MetaOptNet (Lee et al., 2019) | RN18 | Sup. | 64.09 ±0.62 | 80.00 ±0.45 | 65.99 ±0.72 | 81.56 ±0.53 |
| Transductive CNAPS (Bateni et al., 2022) | RN18 | Sup. | 55.60 ±0.90 | 73.10 ±0.70 | 65.90 ±1.10 | 81.80 ±0.70 |
| **BECLR (Ours)** | RN18 | Unsup. | **75.74** ±0.62 | **84.93** ±0.33 | **76.44** ±0.66 | **84.85** ±0.37 |
| PDA-Net (Chen et al., 2021a) | RN50 | Unsup. | 63.84 ±0.91 | 83.11 ±0.56 | 69.01 ±0.93 | 84.20 ±0.69 |
| UniSiam + dist (Lu et al., 2022) | RN50 | Unsup. | 65.33 ±0.36 | 83.22 ±0.24 | 69.60 ±0.38 | 86.51 ±0.26 |
| Meta-DM + UniSiam + dist[*] (Hu et al., 2023a) | RN50 | Unsup. | 66.68 ±0.36 | 85.29 ±0.23 | 69.61 ±0.38 | 86.53 ±0.26 |
| **BECLR (Ours)** | RN50 | Unsup. | **80.57** ±0.57 | **87.82** ±0.29 | **81.69** ±0.61 | **87.86** ±0.32 |

Table 3: Accuracies (in % ± std) on miniImageNet → CDFSL. [†]: our reproduc. Style: **best** and second best.

| Method | ChestX | | ISIC | | EuroSAT | | CropDiseases | |
|---|---|---|---|---|---|---|---|---|
| | 5 way 5-shot | 5 way 20-shot | 5 way 5-shot | 5 way 20-shot | 5 way 5-shot | 5 way 20-shot | 5 way 5-shot | 5 way 20-shot |
| SwAV[†] (Caron et al., 2020) | 25.70 ±0.28 | 30.41 ±0.25 | 40.69 ±0.34 | 49.03 ±0.30 | 84.82 ±0.24 | 90.77 ±0.26 | 88.64 ±0.26 | 95.11 ±0.21 |
| NNCLR[†] (Dwibedi et al., 2021) | 25.74 ±0.41 | 29.54 ±0.45 | 38.85 ±0.56 | 47.82 ±0.53 | 83.45 ±0.57 | 90.80 ±0.39 | 90.76 ±0.57 | 95.37 ±0.37 |
| SAMPTransfer (Shirekar et al., 2023) | 26.27 ±0.44 | **34.15** ±0.50 | 47.60 ±0.59 | **61.28** ±0.56 | 85.55 ±0.60 | 88.52 ±0.50 | 91.74 ±0.55 | 96.36 ±0.28 |
| PsCo (Jang et al., 2022) | 24.78 ±0.23 | 27.69 ±0.23 | 44.00 ±0.30 | 54.59 ±0.29 | 81.08 ±0.35 | 87.65 ±0.28 | 88.24 ±0.31 | 94.95 ±0.18 |
| UniSiam + dist (Lu et al., 2022) | **28.18** ±0.45 | 34.58 ±0.46 | 45.65 ±0.58 | 56.54 ±0.5 | 86.53 ±0.47 | 93.24 ±0.30 | 92.05 ±0.50 | 96.83 ±0.27 |
| ConFeSS (Das et al., 2021) | 27.09 | 33.57 | **48.85** | 60.10 | 84.65 | 90.40 | 88.88 | 95.34 |
| **BECLR (Ours)** | 28.46 ±0.23 | **34.21** ±0.25 | 44.48 ±0.31 | 56.89 ±0.29 | **88.55** ±0.23 | **93.92** ±0.14 | **93.65** ±0.25 | **97.72** ±0.13 |

curated versions of CIFAR-100 (Krizhevsky et al., 2009) for FSL, i.e., CIFAR-FS and FC100. Next, we evaluate BECLR in cross-domain settings on the Caltech-UCSD Birds (CUB) dataset (Welinder et al., 2010) and a more recent cross-domain FSL (CDFSL) benchmark (Guo et al., 2020). For cross-domain experiments, miniImageNet is used as the pretraining (source) dataset and ChestX (Wang et al., 2017), ISIC (Codella et al., 2019), EuroSAT (Helber et al., 2019) and CropDiseases (Mohanty et al., 2016) (in Table 3) and CUB (in Table 12 in the Appendix), as the inference (target) datasets.

## 5.1 EVALUATION RESULTS

We report test accuracies with 95% confidence intervals over 2000 test episodes, each with $Q = 15$ query shots per class, for all datasets, as is most commonly adopted in the literature (Chen et al., 2021a; 2022; Lu et al., 2022). The performance on miniImageNet, tieredImageNet, CIFAR-FS, FC100 and miniImageNet → CUB is evaluated on (5-way, $\{1, 5\}$-shot) classification tasks, whereas for miniImageNet → CDFSL we test on (5-way, $\{5, 20\}$-shot) tasks, as is customary across the literature (Guo et al., 2020; Ericsson et al., 2021). We assess BECLR's performance against a wide variety of baselines ranging from (i) established SSL baselines (Chen et al., 2020a; Grill et al., 2020; Caron et al., 2020; Zbontar et al., 2021; Chen & He, 2021; Dwibedi et al., 2021) to (ii) state-of-the-art U-FSL approaches (Chen et al., 2021a; Lu et al., 2022; Shirekar et al., 2023; Chen et al., 2022; Hu et al., 2023a; Jang et al., 2022), as well as (iii) against a set of competitive supervised baselines (Rusu et al., 2018; Gidaris et al., 2019; Lee et al., 2019; Bateni et al., 2022).

**[A1-a] In-Domain Setting.** The results for miniImageNet and tieredImageNet in the (5-way, $\{1, 5\}$-shot) settings are reported in Table 1. Regardless of backbone depth, BECLR sets a *new state-of-the-art* on both datasets, showing up to a 14% and 2.5% gain on miniImageNet over the prior art of U-FSL for the 1 and 5-shot settings, respectively. The results on tiered-

Table 2: Accuracies in (% ± std) on CIFAR-FS and FC100 in (5-way, $\{1, 5\}$-shot). Style: **best** and second best.

| Method | CIFAR-FS | | FC100 | |
|---|---|---|---|---|
| | 1-shot | 5-shot | 1-shot | 5-shot |
| SimCLR (Chen et al., 2020a) | 54.56 ±0.19 | 71.19 ±0.18 | 36.20 ±0.70 | 49.90 ±0.70 |
| MoCo v2 (Chen et al., 2020b) | 52.73 ±0.20 | 67.81 ±0.19 | 37.70 ±0.70 | 53.20 ±0.70 |
| LF2CS (Li et al., 2022) | 55.04 ±0.72 | 70.62 ±0.57 | 37.20 ±0.70 | 52.80 ±0.60 |
| Barlow Twins (Zbontar et al., 2021) | - | - | 37.90 ±0.70 | 54.10 ±0.60 |
| HMS (Ye et al., 2022) | 54.65 ±0.20 | 73.70 ±0.18 | 37.88 ±0.16 | 53.68 ±0.18 |
| Deep Eigenmaps (Chen et al., 2022) | - | - | 39.70 ±0.70 | 57.90 ±0.70 |
| **BECLR (Ours)** | **70.39** ±0.62 | **81.56** ±0.39 | **45.21** ±0.50 | **60.02** ±0.43 |

ImageNet also highlight a considerable performance margin. Interestingly, BECLR even outperforms U-FSL baselines trained with extra (synthetic) training data, sometimes distilled from deeper backbones, also the cited supervised baselines. Table 2 provides further insights on (the less commonly adopted) CIFAR-FS and FC100 benchmarks, showing a similar trend with up to 15% and 8% in 1 and 5-shot settings, respectively, for CIFAR-FS, and 5.5% and 2% for FC100.

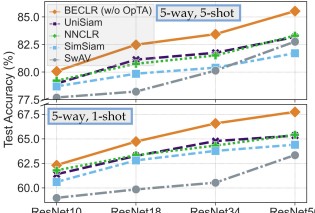
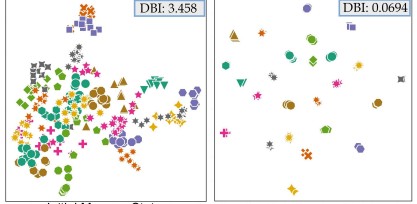
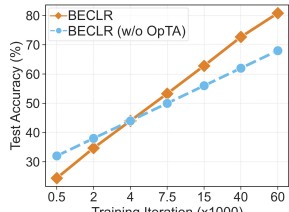

Figure 5: BECLR outperforms all baselines, in terms of U-FSL performance on miniImageNet, even without OpTA.

Figure 6: The dynamic updates of DyCE allow the memory to evolve into a highly separable partitioned latent space. Clusters are denoted by different (colors, markers).

Figure 7: OpTA produces an increasingly larger performance boost as the pretraining feature quality increases.

[**A1-b**] **Cross-Domain Setting.** Following Guo et al. (2020), we pretrain on miniImageNet and evaluate on CDFSL, the results of which are summarized in Tables 3. BECLR again sets a new state-of-the-art on ChestX, EuroSAT, and CropDiseases, and remains competitive on ISIC. Notably, the data distributions of ChestX and ISIC are considerably different from that of miniImageNet. We argue that this influences the embedding quality for the downstream inference, and thus, the efficacy of OpTA in addressing sample bias. Extended versions of Tables 1-3 are found in Appendix C.

[**A1-c**] **Pure Pretraining and OpTA.** To substantiate the impact of the design choices in BECLR, we compare against some of the most influential contrastive SSL approaches: SwAV, SimSiam, NNCLR, and the prior U-FSL state-of-the-art: UniSiam (Lu et al.,

Table 4: BECLR outperforms enhanced prior art with OpTA.

| | miniImageNet | | tieredImageNet | |
|---|---|---|---|---|
| Method | 1 shot | 5 shot | 1 shot | 5 shot |
| CPNWCP+OpTA (Wang et al., 2022a) | 60.45 ±0.81 | 75.84 ±0.56 | 55.05 ±0.31 | 72.91 ±0.26 |
| HMS+OpTA (Ye et al., 2022) | 69.85 ±0.42 | 80.77 ±0.35 | 71.75 ±0.43 | 81.32 ±0.34 |
| PsCo+OpTA (Jang et al., 2022) | 52.89 ±0.71 | 67.42 ±0.54 | 57.46 ±0.59 | 70.70 ±0.45 |
| UniSiam+OpTA (Lu et al., 2022) | 72.54 ±0.61 | 82.46 ±0.32 | 73.37 ±0.64 | 73.37 ±0.64 |
| **BECLR (Ours)** | **75.74 ±0.62** | **84.93 ±0.33** | **76.44 ±0.66** | **84.85 ±0.37** |

2022), in terms of pure pretraining performance, by directly evaluating the pretrained model on downstream FSL tasks (i.e., no OpTA and no fine-tuning). Fig. 5 summarizes this comparison for various network depths in the (5-way, $\{1,5\}$-shot) settings on miniImageNet. BECLR again outperforms all U-FSL/SSL frameworks for all backbone configurations, even without OpTA. As an additional study, in Table 4 we take the opposite steps by plugging in OpTA on a suite of recent prior art in U-FSL. The results demonstrate two important points: (i) OpTA is in fact agnostic to the choice of pretraining method, having considerable impact on downstream performance, and (ii) there still exists a margin between enhanced prior art and BECLR, corroborating that it is not just OpTA that has a meaningful effect but also DyCE and our pretraining methodology.

[**A2**] **Latent Memory Space Evolution.** As a qualitative demonstration, we visualize 30 memory embeddings from 25 partitions $\mathcal{P}_i$ within DyCE for the initial (left) and final (right) state of the latent memory space ($\mathcal{M}$). The 2-D UMAP plots in Fig. 6 provide qualitative evidence of a significant improvement in terms of cluster separation, as training progresses. To quantitatively substantiate this finding, the quality of the memory clusters is also measured by the DBI score (Davies & Bouldin, 1979), with a lower DBI indicating better inter-cluster separation and intra-cluster tightness. The DBI value is significantly lower between partitions $\mathcal{P}_i$ in the final state of $\mathcal{M}$, further corroborating DyCE's ability to establish highly separable and meaningful partitions.

[**A3-a**] **Impact of OpTA.** We visualize the UMAP projections for a randomly sampled (3-way, 1-shot) miniImageNet episode. Fig. 8 illustrates the original $\boldsymbol{P}^{\mathcal{S}}$ (left) and transported $\hat{\boldsymbol{P}}^{\mathcal{S}}$ (right) support prototypes (♦), along with the query set embeddings $\boldsymbol{Z}^{\mathcal{Q}}$ (●) and their kernel density distributions (in contours). As can be seen, the original prototypes are highly biased and deviate from the latent query distributions. OpTA pushes the trans-

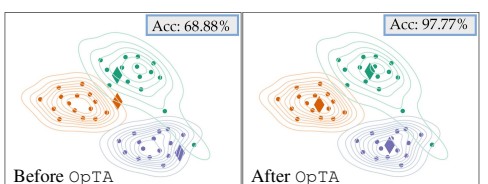

Figure 8: OpTA addresses sample bias, reducing the distribution shift between support and query sets.

ported prototypes much closer to the query distributions (contour centers), effectively diminishing sample bias, resulting in significantly higher accuracy in the corresponding episode.

[**A3-b**] **Relation between Pretraining and Inference.** OpTA operates under the assumption that the query embeddings $\boldsymbol{Z}^{\mathcal{Q}}$ are representative of the actual class distributions. As such, we argue that its efficiency depends on the quality of the extracted features through the pretrained encoder.

Table 6: Hyperparameter ablation study for miniImageNet (5-way, 5-shot) tasks. Accuracies in (% ± std).

| Masking Ratio | | Output Dim. ($d$) | | Neg. Loss Weight ($\lambda$) | | # of NNs ($k$) | | # of Clusters ($P$) | | Memory Size ($M$) | | Memory Module Configuration | |
|---|---|---|---|---|---|---|---|---|---|---|---|---|---|
| Value | Accuracy | Value | Accuracy | Value | Accuracy | Value | Accuracy | Value | Accuracy | Value | Accuracy | Value | Accuracy |
| 10% | 86.59 ±0.25 | 256 | 85.16 ±0.26 | 0.0 | 85.45 ±0.27 | 1 | 86.58 ±0.27 | 100 | 85.27 ±0.24 | 2048 | 85.38 ±0.25 | DyCE-FIFO | 84.05 ±0.39 |
| 30% | **87.82** ±0.29 | 512 | **87.82** ±0.29 | 0.1 | **87.82** ±0.29 | 3 | **87.82** ±0.29 | 200 | **87.82** ±0.29 | 4096 | 86.28 ±0.29 | DyCE-kMeans | 85.37 ±0.33 |
| 50% | 83.36 ±0.28 | 1024 | 85.93 ±0.31 | 0.3 | 86.33 ±0.29 | 5 | 86.79 ±0.26 | 300 | 85.81 ±0.25 | 8192 | **87.82** ±0.29 | DyCE | **87.82** ±0.29 |
| 70% | 77.70 ±0.20 | 2054 | 85.42 ±0.34 | 0.5 | 85.63 ±0.26 | 10 | 86.17 ±0.28 | 500 | 85.45 ±0.20 | 12288 | 85.84 ±0.22 | | |

Fig. 7 assesses this hypothesis by comparing pure pretraining (i.e., BECLR without OpTA) and downstream performance on miniImageNet for the (5-way, 1-shot) setting as training progresses. As can be seen, when the initial pretraining performance is poor, OpTA even leads to performance degradation. On the contrary, it offers an increasingly larger boost as pretraining accuracy improves. The key message here is that these two steps (pretraining and inference) are highly intertwined, further enhancing the overall performance. This notion sits at the core of the BECLR design strategy.

## 5.2 ABLATION STUDIES

**Main Components of BECLR.** Let us investigate the impact of sequentially adding in the four main components of BECLR's end-to-end architecture: (i) masking, (ii) EMA teacher encoder, (iii) DyCE and OpTA. As can be seen from Table 5, when applied individually, masking degrades the performance, but when combined with EMA, it gives a slight boost (1.5%) for both {1, 5}-shot settings.

Table 5: Ablating main components of BECLR.

| Masking | EMA Teacher | DyCE | OpTA | 5-way 1-shot | 5-way 5-shot |
|---|---|---|---|---|---|
| - | - | - | - | 63.57 ±0.43 | 81.42 ±0.28 |
| ✓ | - | - | - | 54.53 ±0.42 | 68.35 ±0.27 |
| - | ✓ | - | - | 65.02 ±0.41 | 82.33 ±0.25 |
| ✓ | ✓ | - | - | 65.33 ±0.44 | 82.69 ±0.26 |
| ✓ | ✓ | ✓ | - | 67.75 ±0.43 | 85.53 ±0.27 |
| ✓ | ✓ | ✓ | ✓ | **80.57** ±0.57 | **87.82** ±0.29 |

DyCE and OpTA are the most crucial components contributing to the overall performance of BECLR. DyCE offers an extra 2.4% and 2.8% accuracy boost in the 1-shot and 5-shot settings, respectively, and OpTA provides another 12.8% and 2.3% performance gain, in the aforementioned settings. As discussed earlier, also illustrated in Fig. 7, the gain of OpTA is owing and proportional to the performance of DyCE. This boost is paramount in the 1-shot scenario where the sample bias is severe.

**Other Hyperparameters.** In Table 6, we summarize the result of ablations on: (i) the masking ratio of student images, (ii) the embedding latent dimension $d$, (iii) the loss weighting hyperparameter $\lambda$ (in Eq. 3), and regarding DyCE: (iv) the number of nearest neighbors selected $k$, (v) the number of memory partitions/clusters $P$, (vi) the size of the memory $M$, and (vii) different memory module configurations. As can be seen, a random masking ratio of 30% yields the best performance. We find that $d = 512$ gives the best results, which is consistent with the literature (Grill et al., 2020; Chen & He, 2021). The negative loss term turns out to be unnecessary to prevent representation collapse (as can be seen when $\lambda = 0$) and $\lambda = 0.1$ yields the best performance. Regarding the number of neighbors selected ($k$), there seems to be a sweet spot in this setting around $k = 3$, where increasing further would lead to inclusion of potentially false positives, and thus performance degradation. $P$ and $M$ are tunable hyperparameters (per dataset) that return the best performance at $P = 200$ and $M = 8192$ in this setting. Increasing $P$, $M$ beyond a certain threshold appears to have a negative impact on cluster formation. We argue that extremely large memory would result in accumulating old embeddings, which might no longer be a good representative of their corresponding classes. Finally, we compare DyCE against two *degenerate* versions of itself: DyCE-FIFO, where the memory has no partitions and is updated with a first-in-first-out strategy; here for incoming embeddings we pick the closest $k$ neighbors. DyCE-kMeans, where we preserve the memory structure and only replace optimal transport with kMeans (with $P$ clusters), in line 8 of Algorithm 2. The performance drop in both cases confirms the importance of the proposed mechanics within DyCE.

## 6 CONCLUDING REMARKS AND BROADER IMPACT

In this paper, we have articulated two key shortcomings of the prior art in U-FSL, to address each of which we have proposed a novel solution embedded within the proposed end-to-end approach, BECLR. The first angle of novelty in BECLR is its dynamic clustered memory module (coined as DyCE) to enhance positive sampling in contrastive learning. The second angle of novelty is an efficient distribution alignment strategy (called OpTA) to address the inherent sample bias in (U-)FSL. Even though tailored towards U-FSL, we believe DyCE has potential *broader impact* on generic self-supervised learning state-of-the-art, as we already demonstrate (in Section 5) that even with OpTA unplugged, DyCE alone empowers BECLR to still outperform the likes of SwaV, SimSiam and NNCLR. OpTA, on the other hand, is an efficient add-on module, which we argue has to become an *integral part* of all (U-)FSL approaches, especially in the more challenging low-shot scenarios.

REPRODUCIBILITY STATEMENT.

To help readers reproduce our experiments, we provide extensive descriptions of implementation details and algorithms. Architectural and training details are provided in Appendices A.1 and A.2, respectively, along with information on the applied data augmentations (Appendix A.3) and tested benchmark datasets (Appendix B.1). The algorithms for BECLR pretraining and DyCE are also provided in both algorithmic (in Algorithms 1, 2) and Pytorch-like pseudocode formats (in Algorithms 3, 4). We have taken every measure to ensure fairness in our comparisons by following the most commonly adopted pretraining and evaluation settings in the U-FSL literature in terms of: pretraining/inference benchmark datasets used for both in-domain and cross-domain experiments, pretraining data augmentations, ($N$-way, $K$-shot) inference settings and number of query set images per tested episode. We also draw baseline results from their corresponding original papers and compare their performance with BECLR for identical backbone depths. For our reproductions (denoted as $^\dagger$) of SwAV and NNCLR we follow the codebase of the original work and adopt it in the U-FSL setup, by following the same augmentations, backbones, and evaluation settings as BECLR. Our codebase is also provided as part of our supplementary material, in an anonymized fashion, and will be made publicly available upon acceptance. All training and evaluation experiments are conducted on 2 A40 NVIDIA GPUs.

ETHICS STATEMENT.

We have read the ICLR Code of Ethics (https://iclr.cc/public/CodeOfEthics) and ensured that this work adheres to it. All benchmark datasets and pretrained model checkpoints are publicly available and not directly subject to ethical concerns.

ACKNOWLEDGEMENTS.

The authors would like to thank Marcel Reinders, Ojas Shirekar and Holger Caesar for insightful conversations and brainstorming. This work was in part supported by Shell.ai Innovation program.

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

# A  IMPLEMENTATION DETAILS

This section describes the implementation and training details of BECLR.

## A.1  ARCHITECTURE DETAILS

BECLR is implemented on PyTorch (Paszke et al., 2019). We use the ResNet family (He et al., 2016) for our backbone networks ($f_\theta$, $f_\psi$). The projection ($g_\theta$, $g_\psi$) and prediction ($h_\theta$) heads are 3- and 2-layer MLPs, respectively, as in Chen & He (2021). Batch normalization (BN) and a ReLU activation function are applied to each MLP layer, except for the output layers. No ReLU activation is applied on the output layer of the projection heads ($g_\theta$, $g_\psi$), while neither BN nor a ReLU activation is applied on the output layer of the prediction head ($h_\theta$). We use a resolution of $224 \times 224$ for input images and a latent embedding dimension of $d = 512$ in all models and experiments, unless otherwise stated. The DyCE memory module consists of a memory unit $\mathcal{M}$, initialized as a random table (of size $M \times d$). We also maintain up to $P$ partitions in $\mathcal{M} = [\mathcal{P}_1, \ldots, \mathcal{P}_P]$, each of which is represented by a prototype stored in $\mathbf{\Gamma} = [\gamma_1, \ldots, \gamma_P]$. Prototypes $\gamma_i$ are the average of the latent embeddings stored in partition $\mathcal{P}_i$. When training on miniImageNet, CIFAR-FS and FC100, we use a memory of size $M = 8192$ that contains $P = 200$ partitions and cluster prototypes ($M = 40960$, $P = 1000$ for tieredImageNet). Note that both $M$ and $P$ are important hyperparameters, whose values were selected by evaluating on the validation set of each dataset for model selection. These hyperparameters would also need to be carefully tuned on an unknown unlabeled training dataset.

## A.2  TRAINING DETAILS

BECLR is pretrained on the training splits of miniImageNet, CIFAR-FS, FC100 and tieredImageNet. We use a batch size of $B = 256$ images for all datasets, except for tieredImageNet ($B = 512$). Following Chen & He (2021), images are resized to $224 \times 224$ for all configurations. We use the SGD optimizer with a weight decay of $10^{-4}$, a momentum of $0.995$, and a cosine decay schedule of the learning rate. Note that we do not require large-batch optimizers, such as LARS (You et al., 2017), or early stopping. Similarly to Lu et al. (2022), the initial learning rate is set to $0.3$ for the smaller miniImageNet, CIFAR-FS, FC100 datasets and $0.1$ for tieredImageNet, and we train for 400 and 200 epochs, respectively. The temperature scalar in the loss function is set to $\tau = 2$. Upon finishing unsupervised pretraining, we only keep the last epoch checkpoint of the student encoder ($f_\theta$) for the subsequent inference stage. For the inference and downstream few-shot classification stage, we create ($N$-way, $K$-shot) tasks from the validation and testing splits of each dataset for model selection and evaluation, respectively. In the inference stage, we sequentially perform up to $\delta \leq 5$ consecutive passes of OpTA, with the transported prototypes of each pass acting as the input of the next pass. The optimal value for $\delta$ for each dataset and ($N$-way, $K$-shot) setting is selected by evaluating on the validation dataset.

Note that at the beginning of pretraining both the encoder representations and the memory embedding space within DyCE are highly volatile. Thus, we allow for an adaptation period epoch < $\text{epoch}_{\text{thr}}$ (empirically 20-50 epochs), during which the batch enhancement path of DyCE is not activated and the encoder is trained via standard contrastive learning (without enhancing the batch with additional positives). On the contrary, the memory updating path of DyCE is activated for every training batch from the beginning of training, allowing the memory to reach a highly separable converged state (see Fig. 6), before plugging in the batch enhancement path in the BECLR pipeline. When the memory space ($\mathcal{M}$) is full for the first time, a kmeans (Likas et al., 2003) clustering step is performed for initializing the cluster prototypes ($\gamma_i$) and memory partitions ($\mathcal{P}_i$). This kmeans clustering step is performed only once during training to initialize the memory prototypes, which are then dynamically updated for each training batch by the memory updating path of DyCE.

## A.3  IMAGE AUGMENTATIONS

The data augmentations that were applied in the pretraining stage of BECLR are showcased in Table 7. These augmentations were applied on the input images for all training datasets. The default data augmentation profile follows a common data augmentation strategy in SSL, including RandomResizedCrop (with scale in [0.2, 1.0]), random ColorJitter (Wu et al., 2018) of {brightness, contrast, saturation, hue} with a probability of 0.1, RandomGrayScale with a probability of 0.2, random GaussianBlur with a probability of 0.5 and a Gaussian kernel in [0.1, 2.0], and finally, RandomHorizontalFlip with a probability of 0.5. Following Lu et al. (2022), this profile can be expanded to

Table 7: Pytorch-like descriptions of the data augmentation profiles applied on the pretraining phase of `BECLR`.

| Data Augmentation Profile | Description |
|---|---|
| **Default** | RandomResizedCrop(size=224, scale=(0.2, 1))
RandomApply([ColorJitter(brightness=0.4, contrast=0.4, saturation=0.4, hue=0.1)], p=0.1)
RandomGrayScale(p=0.2)
RandomApply([GaussianBlur([0.1, 2.0])], p=0.5)
RandomHorizontalFlip(p=0.5) |
| **Strong** | RandomResizedCrop(size=224, scale=(0.2, 1))
RandomApply([ColorJitter(brightness=0.4, contrast=0.4, saturation=0.2, hue=0.1)], p=0.1)
RandomGrayScale(p=0.2)
RandomApply([GaussianBlur([0.1, 2.0])], p=0.5)
RandAugment(n=2, m=10, mstd=0.5)
RandomHorizontalFlip(p=0.5)
RandomVerticalFlip(p=0.5) |

the strong data augmentation profile, which also includes RadomVerticalFlip with a probability of $0.5$ and RandAugment (Cubuk et al., 2020) with $n = 2$ layers, a magnitude of $m = 10$, and a noise of the standard deviation of magnitude of $mstd = 0.5$. Unless otherwise stated, the strong data augmentation profile is applied on all training images before being passed to the backbone encoders.

## B  EXPERIMENTAL SETUP

In this section, we provide more detailed information regarding all the few-shot benchmark datasets that were used as part of our experimental evaluation (in Section 5), along with `BECLR`'s training and evaluation procedures for both in-domain and cross-domain U-FSL settings, for ensuring fair comparisons with U-FSL baselines and our reproductions (for SwAV and NNCLR).

Table 8: Overview of cross-domain few-shot benchmarks, on which `BECLR` is evaluated. The datasets are sorted with decreasing (distribution) domain similarity to ImageNet and miniImageNet.

| ImageNet similarity | Dataset | # of classes | # of samples |
|---|---|---|---|
| High | CUB (Welinder et al., 2010) | 200 | 11,788 |
| Low | CropDiseases (Mohanty et al., 2016) | 38 | 43,456 |
| Low | EuroSAT (Helber et al., 2019) | 10 | 27,000 |
| Low | ISIC (Codella et al., 2019) | 7 | 10,015 |
| Low | ChestX (Wang et al., 2017) | 7 | 25,848 |

### B.1  DATASET DETAILS

**miniImageNet.** It is a subset of ImageNet (Russakovsky et al., 2015), containing 100 classes with 600 images per class. We randomly select 64, 16, and 20 classes for training, validation, and testing, following the predominantly adopted settings of Ravi & Larochelle (2016).

**tieredImageNet.** It is a larger subset of ImageNet, containing 608 classes and a total of $779,165$ images, grouped into 34 high-level categories, 20 (351 classes) of which are used for training, 6 (97 classes) for validation and 8 (160 classes) for testing.

**CIFAR-FS.** It is a subset of CIFAR-100 (Krizhevsky et al., 2009), which is focused on FSL tasks by following the same sampling criteria that were used for creating miniImageNet. It contains 100 classes with 600 images per class, grouped into 64, 16, 20 classes for training, validation, and testing, respectively. The additional challenge here is the limited original image resolution of $32 \times 32$.

**FC100.** It is also a subset of CIFAR-100 (Krizhevsky et al., 2009) and contains the same 60000 ($32 \times 32$) images as CIFAR-FS. Here, the original 100 classes are grouped into 20 superclasses, in such a way as to minimize the information overlap between training, validation and testing classes (McAllester & Stratos, 2020). This makes this data set more demanding for (U-)FSL, since training and testing classes are highly dissimilar. The training split contains 12 superclasses (of 60 classes), while both the validation and testing splits are composed of 4 superclasses (of 20 classes).

**CDFSL.** It consists of four distinct datasets with decreasing domain similarity to ImageNet, and by extension miniImageNet, ranging from crop disease images in CropDiseases (Mohanty et al., 2016) and aerial satellite images in EuroSAT (Helber et al., 2019) to dermatological skin lesion images in ISIC2018 (Codella et al., 2019) and grayscale chest X-ray images in ChestX (Wang et al., 2017).

**CUB.** It consists of 200 classes and a total of 11,788 images, split into 100 classes for training and 50 for both validation and testing, following the split settings of Chen et al. (2018). Additional information on the cross-domain few-shot benchmarks (CUB and CDFSL) is provided in Table 8.

### B.2 PRETRAINING AND EVALUATION PROCEDURES

For all in-domain experiments BECLR is pretrained on the training split of the selected dataset (miniImageNet, tieredImageNet, CIFAR-FS, or FC100), followed by the subsequent inference stage on the validation and testing splits of the same dataset for model selection and final evaluation, respectively. In contrast, in the cross-domain setting BECLR is pretrained on the training split of miniImageNet and then evaluated on the validation and test splits of either CDFSL (ChestX, ISIC, EuroSAT, CropDiseases) or CUB. We report test accuracies with $95\%$ confidence intervals over 2000 test episodes, each with 15 query shots per class, for all tested datasets, as is most commonly adopted in the literature (Chen et al., 2021a; 2022; Lu et al., 2022). The performance on miniImageNet, tieredImageNet, CIFAR-FS, FC100 and miniImageNet $\rightarrow$ CUB is evaluated on (5-way, $\{1, 5\}$-shot) classification tasks, while on miniImageNet $\rightarrow$ CDFSL we evaluate on (5-way, $\{5, 20\}$-shot) tasks, as is customary across the literature (Guo et al., 2020).

We have taken every measure to ensure fairness in our comparison with U-FSL baselines and our reproductions. To do so, all compared baselines have the same pretraining and testing dataset (in both in-domain and cross-domain scenarios), follow similar data augmentation profiles as part of their pretraining and are evaluated in identical ($N$-way, $K$-shot) FSL settings, on the same number of query set images, for all tested inference episodes. We also draw baseline results from their corresponding original papers and compare their performance with BECLR for identical backbone depths (roughly similar parameter count for all methods). For our reproductions (SwAV and NNCLR), we follow the codebase of the original work and adopt it in the U-FSL setup, by following the same augmentations, backbones, and evaluation settings as BECLR.

## C  EXTENDED RESULTS AND VISUALIZATIONS

This section provides extended experimental findings, both quantitative and qualitative, complementing the experimental evaluation in Section 5 and providing further intuition on BECLR.

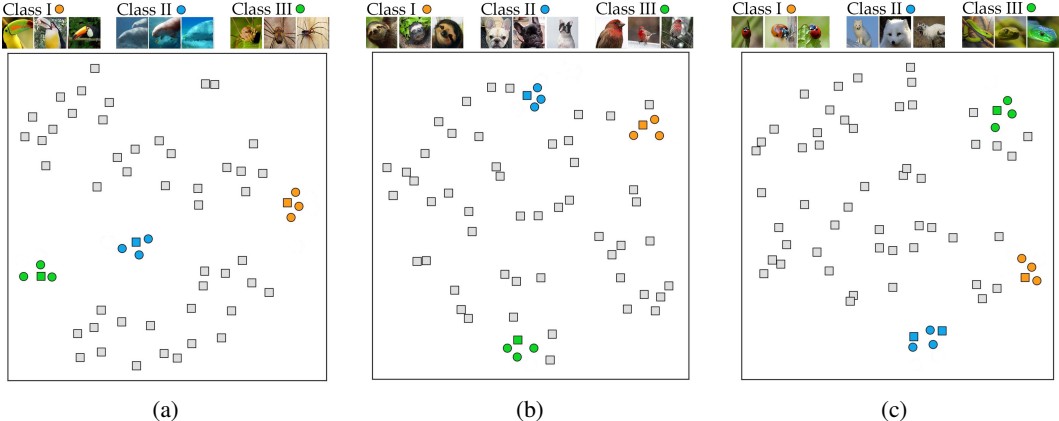

Figure 9: (a, b): All 3 support images (●), for all 3 classes, get assigned to a single class-representative memory prototype (■). (c): A cluster can be split into multiple subclusters, without degrading the model performance. Colors denote the classes of the random (3-way, 3-shot) episode and the assigned memory prototypes.

### C.1  ANALYSIS ON PSEUDOCLASS-COGNIZANCE

As discussed in Section 5, the dynamic equipartitioned updates of DyCE facilitate the creation of a highly separable latent memory space, which is then used for sampling additional meaningful

positive pairs. We argue that these separable clusters and their prototypes capture a semblance of class-cognizance. As a qualitative demonstration, Fig. 9 depicts the 2-D UMAP (McInnes et al., 2018) projections of the support embeddings (●) for 3 (Figs. 9a, 9b, 9c) randomly sampled mini-ImageNet (3-way, 3-shot) tasks, along with 50 randomly selected memory prototypes (□) from the last-epoch checkpoint of DyCE. Only memory prototypes assigned to support embeddings (■) are colored (with the corresponding class color). As can be seen in Figs. 9a, 9b, all 3-shot embeddings for each class are assigned to a single memory prototype, which shows that the corresponding prototype, and by extension memory cluster, is capable of capturing the latent representation of a class even without access to base class labels. Note that as the training progresses, a memory cluster can split into two or more (sub-)clusters, corresponding to the same latent training class. For example, Fig. 9c illustrates such a case where the fox class has been split between two different prototypes. This is to be expected since there is no "one-to-one" relationship between memory clusters and latent training classes (we do not assume any prior knowledge on the classes of the unlabeled pre-training dataset and the number of memory clusters is an important hyperparameter to be tuned). Notably, BECLR still treats images sampled from these two (sub-)clusters as positives, which does not degrade the model's performance assuming that both of these (sub-)clusters are consistent (i.e., contain only fox image embeddings).

Table 9: Extended version of Table 1. Accuracies (in % ± std) on miniImageNet and tieredImageNet compared against unsupervised (Unsup.) and supervised (Sup.) baselines. Encoders: RN: ResNet, Conv: convolutional blocks. †: denotes our reproduction. *: extra synthetic training data. Style: **best** and second best.

| Method | Backbone | Setting | miniImageNet | | tieredImageNet | |
|---|---|---|---|---|---|---|
| | | | 5-way 1-shot | 5-way 5-shot | 5-way 1-shot | 5-way 5-shot |
| ProtoTransfer (Medina et al., 2020) | Conv4 | Unsup. | 45.67 ± 0.79 | 62.99 ± 0.75 | - | - |
| Meta-GMVAE (Lee et al., 2020) | Conv4 | Unsup. | 42.82 ± 0.45 | 55.73 ± 0.39 | - | - |
| C3LR (Shirekar & Jamali-Rad, 2022) | Conv4 | Unsup. | 47.92 ± 1.20 | 64.81 ± 1.15 | 42.37 ± 0.77 | 61.77 ± 0.25 |
| SAMPTransfer (Shirekar et al., 2023) | Conv4b | Unsup. | 61.02 ± 1.05 | 72.52 ± 0.68 | 49.10 ± 0.94 | 65.19 ± 0.82 |
| LF2CS (Li et al., 2022) | RN12 | Unsup. | 47.93 ± 0.19 | 66.44 ± 0.17 | 53.16 ± 0.66 | 66.59 ± 0.57 |
| CPNWCP (Wang et al., 2022a) | RN18 | Unsup. | 53.14 ± 0.62 | 67.36 ± 0.5 | 45.00 ± 0.19 | 62.96 ± 0.19 |
| SimCLR (Chen et al., 2020a) | RN18 | Unsup. | 62.58 ± 0.37 | 79.66 ± 0.27 | 63.38 ± 0.42 | 79.17 ± 0.34 |
| SwAV† (Caron et al., 2020) | RN18 | Unsup. | 59.84 ± 0.52 | 78.23 ± 0.26 | 65.26 ± 0.53 | 81.73 ± 0.24 |
| NNCLR† (Dwibedi et al., 2021) | RN18 | Unsup. | 63.33 ± 0.53 | 80.75 ± 0.25 | 65.46 ± 0.55 | 81.40 ± 0.27 |
| SimSiam (Chen & He, 2021) | RN18 | Unsup. | 62.80 ± 0.37 | 79.85 ± 0.27 | 64.05 ± 0.40 | 81.40 ± 0.30 |
| HMS (Ye et al., 2022) | RN18 | Unsup. | 58.20 ± 0.23 | 75.77 ± 0.16 | 58.42 ± 0.25 | 75.85 ± 0.18 |
| Laplacian Eigenmaps (Chen et al., 2022) | RN18 | Unsup. | 59.47 ± 0.87 | 78.79 ± 0.58 | - | - |
| PsCo† (Jang et al., 2022) | RN18 | Unsup. | 47.24 ± 0.76 | 65.48 ± 0.68 | 54.33 ± 0.54 | 69.73 ± 0.49 |
| UniSiam + dist (Lu et al., 2022) | RN18 | Unsup. | 64.10 ± 0.36 | 82.26 ± 0.25 | 67.01 ± 0.39 | 84.47 ± 0.28 |
| Meta-DM + UniSiam + dist* (Hu et al., 2023a) | RN18 | Unsup. | 65.64 ± 0.36 | 83.97 ± 0.25 | 67.11 ± 0.40 | 84.39 ± 0.28 |
| MetaOptNet (Lee et al., 2019) | RN18 | Sup. | 64.09 ± 0.62 | 80.00 ± 0.45 | 65.99 ± 0.72 | 81.56 ± 0.53 |
| Transductive CNAPS (Bateni et al., 2022) | RN18 | Sup. | 55.60 ± 0.90 | 73.10 ± 0.70 | 65.90 ± 1.10 | 81.80 ± 0.70 |
| MAML (Finn et al., 2017) | RN34 | Sup. | 51.46 ± 0.90 | 65.90 ± 0.79 | 51.67 ± 1.81 | 70.30 ± 1.75 |
| ProtoNet (Snell et al., 2017) | RN34 | Sup. | 53.90 ± 0.83 | 74.65 ± 0.64 | 51.67 ± 1.81 | 70.30 ± 1.75 |
| **BECLR (Ours)** | RN18 | Unsup. | **75.74** ± 0.62 | **84.93** ± 0.33 | **76.44** ± 0.66 | **84.85** ± 0.37 |
| SwAV† (Caron et al., 2020) | RN50 | Unsup. | 63.34 ± 0.42 | 82.76 ± 0.24 | 68.02 ± 0.52 | 85.93 ± 0.33 |
| NNCLR† (Dwibedi et al., 2021) | RN50 | Unsup. | 65.42 ± 0.44 | 83.31 ± 0.21 | 69.82 ± 0.54 | 86.41 ± 0.31 |
| TrainProto (Li & Liu, 2021) | RN50 | Unsup. | 58.92 ± 0.91 | 73.94 ± 0.63 | - | - |
| UBC-FSL (Chen et al., 2021b) | RN50 | Unsup. | 56.20 ± 0.60 | 75.40 ± 0.40 | 66.60 ± 0.70 | 83.10 ± 0.50 |
| PDA-Net (Chen et al., 2021a) | RN50 | Unsup. | 63.84 ± 0.91 | 83.11 ± 0.56 | 69.01 ± 0.93 | 84.20 ± 0.69 |
| UniSiam + dist (Lu et al., 2022) | RN50 | Unsup. | 65.33 ± 0.36 | 83.22 ± 0.24 | 69.60 ± 0.38 | 86.51 ± 0.26 |
| Meta-DM + UniSiam + dist* (Hu et al., 2023a) | RN50 | Unsup. | 66.68 ± 0.36 | 85.29 ± 0.23 | 69.61 ± 0.38 | 86.53 ± 0.26 |
| **BECLR (Ours)** | RN50 | Unsup. | **80.57** ± 0.57 | **87.82** ± 0.29 | **81.69** ± 0.61 | **87.86** ± 0.32 |

Table 10: Extended version of Table 2. Accuracies in (% ± std) on CIFAR-FS and FC100 of different U-FSL baselines. Encoders: RN: ResNet. †: denotes our reproduction. Style: **best** and second best.

| Method | Backbone | Setting | CIFAR-FS | | FC100 | |
|---|---|---|---|---|---|---|
| | | | 5-way 1-shot | 5-way 5-shot | 5-way 1-shot | 5-way 5-shot |
| Meta-GMVAE (Lee et al., 2020) | RN18 | Unsup. | - | - | 36.30 ± 0.70 | 49.70 ± 0.80 |
| SimCLR (Chen et al., 2020a) | RN18 | Unsup. | 54.56 ± 0.19 | 71.19 ± 0.18 | 36.20 ± 0.70 | 49.90 ± 0.70 |
| MoCo v2 (Chen et al., 2020b) | RN18 | Unsup. | 52.73 ± 0.20 | 67.81 ± 0.19 | 37.70 ± 0.70 | 53.20 ± 0.70 |
| MoCHi (Kalantidis et al., 2020) | RN18 | Unsup. | 50.42 ± 0.22 | 65.91 ± 0.20 | 37.51 ± 0.17 | 48.95 ± 0.17 |
| BYOL (Grill et al., 2020) | RN18 | Unsup. | 51.33 ± 0.21 | 66.73 ± 0.18 | 37.20 ± 0.70 | 52.80 ± 0.60 |
| LF2CS (Li et al., 2022) | RN18 | Unsup. | 55.04 ± 0.72 | 70.62 ± 0.57 | 37.20 ± 0.70 | 52.80 ± 0.60 |
| CUMCA (Xu et al., 2021) | RN18 | Unsup. | 50.48 ± 0.12 | 67.83 ± 0.18 | 33.00 ± 0.17 | 47.41 ± 0.19 |
| Barlow Twins (Zbontar et al., 2021) | RN18 | Unsup. | - | - | 37.90 ± 0.70 | 54.10 ± 0.60 |
| HMS (Ye et al., 2022) | RN18 | Unsup. | 54.65 ± 0.20 | 73.70 ± 0.18 | 37.88 ± 0.16 | 53.68 ± 0.18 |
| Deep Eigenmaps (Chen et al., 2022) | RN18 | Unsup. | - | - | 39.70 ± 0.70 | 57.90 ± 0.70 |
| **BECLR (Ours)** | RN18 | Unsup. | **70.39** ± 0.62 | **81.56** ± 0.39 | **45.21** ± 0.50 | **60.02** ± 0.43 |

Table 11: Extended version of Table 3. Accuracies (in % ± std) on miniImageNet → CDFSL. $^{\dagger}$: denotes our reproduction. Style: **best** and second best.

| Method | ChestX | | ISIC | | EuroSAT | | CropDiseases | |
|---|---|---|---|---|---|---|---|---|
| | 5 way 5-shot | 5 way 20-shot | 5 way 5-shot | 5 way 20-shot | 5 way 5-shot | 5 way 20-shot | 5 way 5-shot | 5 way 20-shot |
| ProtoTransfer (Medina et al., 2020) | 26.71 ±0.46 | 33.82 ±0.48 | 45.19 ±0.56 | 59.07 ±0.55 | 75.62 ±0.67 | 86.80 ±0.42 | 86.53 ±0.56 | 95.06 ±0.32 |
| BYOL (Grill et al., 2020) | 26.39 ±0.43 | 30.71 ±0.47 | 43.09 ±0.56 | 53.76 ±0.55 | 83.64 ±0.54 | 89.62 ±0.39 | 92.71 ±0.47 | 96.07 ±0.33 |
| MoCo v2 (Chen et al., 2020b) | 25.26 ±0.44 | 29.43 ±0.46 | 42.60 ±0.55 | 52.39 ±0.49 | 84.15 ±0.52 | 88.92 ±0.41 | 87.62 ±0.60 | 92.12 ±0.46 |
| SwAV$^{\dagger}$ (Caron et al., 2020) | 25.70 ±0.28 | 30.41 ±0.25 | 40.69 ±0.34 | 49.03 ±0.30 | 84.82 ±0.24 | 90.77 ±0.26 | 88.64 ±0.26 | 95.11 ±0.21 |
| SimCLR (Chen et al., 2020a) | 26.36 ±0.44 | 30.82 ±0.43 | 43.99 ±0.55 | 53.00 ±0.54 | 82.78 ±0.56 | 89.38 ±0.40 | 90.29 ±0.52 | 94.03 ±0.37 |
| NNCLR$^{\dagger}$ (Dwibedi et al., 2021) | 25.74 ±0.41 | 29.54 ±0.45 | 38.85 ±0.56 | 47.82 ±0.53 | 83.45 ±0.57 | 90.80 ±0.39 | 90.76 ±0.57 | 95.37 ±0.37 |
| C3LR (Shirekar & Jamali-Rad, 2022) | 26.00 ±0.41 | 33.39 ±0.47 | 45.93 ±0.54 | 59.95 ±0.53 | 80.32 ±0.65 | 88.09 ±0.45 | 87.90 ±0.55 | 95.38 ±0.31 |
| SAMPTransfer (Shirekar et al., 2023) | 26.27 ±0.44 | 34.15 ±0.50 | 47.60 ±0.59 | **61.28** ±0.56 | 85.55 ±0.60 | 88.52 ±0.50 | 91.74 ±0.55 | 96.36 ±0.28 |
| PsCo (Jang et al., 2022) | 24.78 ±0.23 | 27.69 ±0.23 | 44.00 ±0.30 | 54.59 ±0.29 | 81.08 ±0.35 | 87.65 ±0.28 | 88.24 ±0.31 | 94.95 ±0.18 |
| UniSiam + dist (Lu et al., 2022) | **28.18** ±0.45 | **34.58** ±0.46 | 45.65 ±0.58 | 56.54 ±0.45 | 86.53 ±0.47 | 93.24 ±0.30 | 92.05 ±0.50 | 96.83 ±0.27 |
| ConFeSS (Das et al., 2021) | 27.09 | 33.57 | **48.85** | 60.10 | 84.65 | 90.40 | 88.88 | 95.34 |
| ATA (Wang & Deng, 2021) | 24.43 ±0.2 | - | 45.83 ±0.3 | - | 83.75 ±0.4 | - | 90.59 ±0.3 | - |
| **BECLR (Ours)** | **28.46** ±0.23 | **34.21** ±0.25 | 44.48 ±0.31 | 56.89 ±0.29 | **88.55** ±0.23 | **93.92** ±0.14 | **93.65** ±0.25 | **97.72** ±0.13 |

## C.2 IN-DOMAIN SETTING

Here, we provide more extensive experimental results on miniImageNet, tieredImageNet, CIFAR-FS and FC100 by comparing against additional baselines. Table 9 corresponds to an extended version of Table 1 and, similarly, Table 10 is an extended version of Table 2. We assess the performance of BECLR against a wide variety of methods: from (i) established SSL baselines (Chen et al., 2020a; Caron et al., 2020; Grill et al., 2020; Kalantidis et al., 2020; Chen et al., 2020b; Zbontar et al., 2021; Xu et al., 2021; Chen & He, 2021; Dwibedi et al., 2021) to (ii) state-of-the-art U-FSL approaches (Hsu et al., 2018; Khodadadeh et al., 2019; Medina et al., 2020; Lee et al., 2020; Chen et al., 2021b; Li & Liu, 2021; Ye et al., 2022; Shirekar & Jamali-Rad, 2022; Li et al., 2022; Wang et al., 2022a; Chen et al., 2022; Lu et al., 2022; Jang et al., 2022; Shirekar et al., 2023; Hu et al., 2023a). Furthermore, we compare with a set of supervised baselines (Finn et al., 2017; Snell et al., 2017; Rusu et al., 2018; Gidaris et al., 2019; Lee et al., 2019; Bateni et al., 2022).

## C.3 CROSS-DOMAIN SETTING

We also provide an extended version of the experimental results in the miniImageNet → CDFSL cross-domain setting of Table 3, where we compare against additional baselines, as seen in Table 11. Additionally, in Table 12 we evaluate the performance of BECLR on the miniImageNet → CUB cross-domain setting. We compare against any existing unsupervised baselines (Hsu et al., 2018; Khodadadeh et al., 2019; Chen et al., 2020a; Caron et al., 2020; Medina et al., 2020; Grill et al., 2020; Chen et al., 2020b; Zbontar et al., 2021; Ye et al., 2022; Chen et al., 2022; Lu et al., 2022; Shirekar et al., 2023; Shirekar & Jamali-Rad,

Table 12: Accuracies in (% ± std) on miniImageNet → CUB. $^{\dagger}$: denotes our reproduction. Style: **best** and second best.

| Method | miniImageNet → CUB | |
|---|---|---|
| | 5-way 1-shot | 5-way 5-shot |
| Meta-GMVAE (Lee et al., 2020) | 38.09 ±0.47 | 55.65 ±0.42 |
| SimCLR (Chen et al., 2020a) | 38.25 ±0.49 | 55.89 ±0.46 |
| MoCo v2 (Chen et al., 2020b) | 39.29 ±0.47 | 56.49 ±0.44 |
| BYOL (Grill et al., 2020) | 40.63 ±0.46 | 56.92 ±0.43 |
| SwAV$^{\dagger}$ (Caron et al., 2020) | 38.34 ±0.51 | 53.94 ±0.43 |
| NNCLR$^{\dagger}$ (Dwibedi et al., 2021) | 39.37 ±0.53 | 54.78 ±0.42 |
| Barlow Twins (Zbontar et al., 2021) | 40.46 ±0.47 | 57.16 ±0.42 |
| Laplacian Eigenmaps (Chen et al., 2022) | 41.08 ±0.48 | 58.86 ±0.45 |
| HMS (Ye et al., 2022) | 40.75 | 58.32 |
| PsCo (Jang et al., 2022) | - | 57.38 ±0.44 |
| **BECLR (Ours)** | **43.45** ±0.50 | **59.51** ±0.46 |

2022; Jang et al., 2022; Lee et al., 2020), for which this more challenging cross-domain experiment has been conducted (to the best of our knowledge).

## D COMPLEXITY ANALYSIS

In this section, we analyze the computational and time complexity of BECLR and compare with different contrastive learning baselines, as summarized in Table 13. Neither DyCE nor OpTA introduce additional trainable parameters; thus, the total parameter count of BECLR is on par with standard Siamese architectures and dependent on the backbone configuration. BECLR utilizes a student-teacher EMA architecture, hence needs to store separate weights for 2 distinct networks, denoted as ResNet 2×, similar to BYOL (Grill et al., 2020). A batch size of 256 and 512 is used for training BECLR on miniImageNet and tieredImageNet, respectively, which again is standard practice in the U-FSL literature. However, DyCE artificially enhances the batch, on which the contrastive loss is applied, to $k + 1$ times the size of the original, in effect slightly increasing the training time of BECLR. OpTA also introduces additional calculations in the inference time, which nevertheless results in a negligible increase in terms of the average episode inference time of BECLR.

Table 13: Comparison of the computational complexity of `BECLR` with different contrastive SSL approaches in terms of parameter count, training, and inference times. [†]: denotes our reproduction.

| Method | Backbone Architecture | Backbone Parameter Count (M) | Training Time (sec/epoch) | Inference Time (sec/episode) |
|---|---|---|---|---|
| SwAV[†] (Caron et al., 2020) | ResNet-18 (1×) | 11.2 | 124.51 | 0.213 |
| NNCLR[†] (Dwibedi et al., 2021) | ResNet-18 (2×) | 22.4 | 174.13 | 0.211 |
| UniSiam (Lu et al., 2022) | ResNet-18 (1×) | 11.2 | 153.37 | 0.212 |
| BECLR (Ours) | ResNet-18 (2×) | 22.4 | 190.20 | 0.216 |
| SwAV[†] (Caron et al., 2020) | ResNet-50 (1×) | 23.5 | 136.82 | 0.423 |
| NNCLR[†] (Dwibedi et al., 2021) | ResNet-50 (2×) | 47.0 | 182.33 | 0.415 |
| UniSiam (Lu et al., 2022) | ResNet-50 (1×) | 23.5 | 167.71 | 0.419 |
| BECLR (Ours) | ResNet-50 (2×) | 47.0 | 280.53 | 0.446 |

# E    PSEUDOCODE

This section includes the algorithms for the pretraining methodology of `BECLR` and the proposed dynamic clustered memory (`DyCE`) in a Pytorch-like pseudocode format. Algorithm 3 provides an overview of the pretraining stage of `BECLR` and is equivalent to Algorithm 1, while Algorithm 4 describes the two informational paths of `DyCE` and is equivalent to Algorithm 2.

---

**Algorithm 3:** Unsupervised Pretraining of `BECLR`: PyTorch-like Pseudocode

```
# {f, g, h}_student: student backbone, projector, and predictor
# {f, g}_teacher: teacher backbone and projector
# DyCE_{student, teacher}: our dynamic clustered memory module for student and teacher paths (see Algorithm. 4)
def  BECLR(x):                                                    # x: a random training mini-batch of L samples
    x = [x1,  x2] = [aug1(x),  aug2(x)]                           # concatenate the two augmented views of x
    z_s = h_student( g_student( f_student( mask(x))))             # (2B × d): extract student representations
    z_t = g_teacher( f_teacher(x)).detach()                       # (2B × d): extract teacher representations
    z_s, z_t = DyCE_student(z_s), DyCE_teacher(z_t) # update memory via optimal transport & compute enhanced batch (2B(k+1) × d)
    loss_pos = - (z_s * z_t).sum(dim=1).mean()                         # compute positive loss term
    loss = loss_pos + (matmul(z_s, z_t.T) * mask).div(temp).exp().sum(dim=1)).div(n_neg).mean().log()      # compute final loss term
    loss.backward(),  momentum_update( student.parameters, teacher.parameters)         # update student and teacher parameters
```

---

**Algorithm 4:** Dynamic Clustered Memory (`DyCE`): PyTorch-like Pseudocode

```
# z: batch representations (2B×d)
# self.memory: memory embedding space (M×d)
# self.prototypes: memory partition prototypes (P×d)
def  DyCE(self, z):
  if  self.memory.shape[0] == M:
      # - - - - - Path I: Top-k NNs Selection and Batch Enhancement - - - - -
      if  epoch ≥ epoch_thr
        batch_prototypes = assign_prototypes(z, self.prototypes)        # (2B×d): find nearest memory prototype for each batch embedding
        y_mem = topk(self.memory, z, batch_prototypes)        # (2Bk×d): find top-k NNs, from memory partition of nearest prototype
        z = [z, y_mem]             # (2B(k+1)×d): concatenate batch and memory representations to create the final enhanced batch
      # - - - - - Path II: Iterative Memory Updating - - - - -
      opt_plan = sinkhorn( D(z, self.prototypes))       # get optimal assignments between batch embeddings and prototypes (Solve Eq. 2)
      self.update(z, opt_plan)   # add latest batch to memory and update memory partitions and prototypes, using the optimal assignments
      self.dequeue()                                           # discard the 2B oldest memory embeddings
  else:
      self.enqueue(z)                              # simply store latest batch until the memory is full for the first time
  return  z
```

---

# F    IN-DEPTH COMPARISON WITH PSCO

In this section we perform an comparative analysis of `BECLR` with PsCo (Jang et al., 2022), in terms of their motivation, similarities, discrepancies, and performance.

## F.1    MOTIVATION AND DESIGN CHOICES

`BECLR` and PsCo indeed share some similarities, in that both methods utilize a student-teacher momentum architecture, a memory module of past representations, some form of contrastive loss, and optimal transport (even though for different purposes). Note that none of these components are unique to neither PsCo nor `BECLR`, but can be found in the overall U-FSL and SSL literature (He et al., 2020; Dwibedi et al., 2021; Lu et al., 2022; Wang et al., 2022a; Ye et al., 2022).

Let us now expand on their discrepancies and the unique aspects of `BECLR`: (i) PsCo is based on meta learning, constructing few-shot classification (i.e., $N$-way $K$-shot) tasks, during meta-training and relies on fine-tuning to rapidly adapt to novel tasks during meta-testing. In contrast, `BECLR` is a contrastive framework based on metric/transfer learning, which focuses on representation quality and relies on `OpTA` for transferring to novel tasks (no few-shot tasks during pretraining, no fine-tuning during inference/testing). (ii) PsCo utilizes a simple FIFO memory queue and is oblivious to class-level information, while `BECLR` maintains a clustered highly-separable (as seen in Fig. 6) latent space in `DyCE`, which, after an adaptation period, is used for sampling meaningful positives. (iii) PsCo applies optimal transport for creating pseudolabels (directly from the unstructured memory queue) for $N*K$ support embeddings, in turn used as a supervisory signal to enforce consistency between support (drawn from teacher) and query (student) embeddings of the created pseudolabeled task. In stark contrast, `BECLR` artificially enhances the batch with additional positives and applies an instance-level contrastive loss to enforce consistency between the original and enhanced (additional) positive pairs. After each training iteration, optimal transport is applied to update the stored clusters within `DyCE` in an equipartitioned fashion with embeddings from the current batch. (iv) Finally, `BECLR` also incorporates optimal transport (in `OpTA`) to align the distributions between support and query sets, during inference, which does not share similarity with the end-to-end pipeline of PsCo.

## F.2 PERFORMANCE AND ROBUSTNESS

We conduct an additional experiment, summarized in Table 14, to study the impact of adding components of `BECLR` (symmetric loss, masking, `OpTA`) to PsCo and the robustness of `BECLR`. For the purposes of this experiment, we reproduced PsCo on a deeper ResNet-18 backbone, ensuring fairness in our comparisons. Next, we modified its loss function to be symmetric (allowing both augmentations of $X$ to pass through both branches) similar to `BECLR`, (this is denoted as `PsCo+`). Next, we also add patchwise masking to PsCo (denoted as `PsCo++`) and as an additional step we have also added

Table 14: Accuracies in (% $\pm$ std) on miniImageNet. [†]: denotes our reproduction. Style: **best** and second best.

| Method | Backbone | 5 way 1 shot | 5 way 5 shot |
|---|---|---|---|
| PsCo (Jang et al., 2022) | Conv5 | 46.70 ± 0.42 | 63.26 ± 0.37 |
| PsCo[†] (Jang et al., 2022) | RN18 | 47.24 ± 0.46 | 65.48 ± 0.38 |
| PsCo+[†] (Jang et al., 2022) | RN18 | 47.86 ± 0.44 | 65.95 ± 0.37 |
| PsCo++[†] (Jang et al., 2022) | RN18 | 47.58 ± 0.45 | 65.74 ± 0.38 |
| PsCo w/ OpTA[†] (Jang et al., 2022) | RN18 | 52.89 ± 0.61 | 67.42 ± 0.51 |
| PsCo+ w/ OpTA[†] (Jang et al., 2022) | RN18 | 54.43 ± 0.59 | 68.31 ± 0.52 |
| PsCo++ w/ OpTA[†] (Jang et al., 2022) | RN18 | 54.35 ± 0.60 | 68.43 ± 0.52 |
| BECLR (Ours) | RN18 | **75.74 ± 0.62** | **84.93 ± 0.33** |
| BECLR− (Ours) | RN18 | 74.37 ± 0.61 | 84.19 ± 0.31 |
| BECLR−− (Ours) | RN18 | 73.65 ± 0.61 | 83.63 ± 0.31 |
| BECLR w/o OpTA (Ours) | RN18 | 66.14 ± 0.43 | 84.32 ± 0.27 |
| BECLR− w/o OpTA (Ours) | RN18 | 65.26 ± 0.41 | 83.68 ± 0.25 |
| BECLR−− w/o OpTA (Ours) | RN18 | 64.68 ± 0.44 | 83.45 ± 0.26 |

our novel module `OpTA` on top of all these models (`PsCo w/ OpTA`), during inference. We observe that neither the symmetric loss nor masking offer meaningful improvements in the performance of PsCo. On the contrary, `OpTA` yields a significant performance boost (up to $6.5\%$ in the 1-shot setting, in which the sample bias is most severe). This corroborates our claim that our proposed `OpTA` should be considered as an add-on module to every (U-)FSL approach out there.

As an additional robustness study we take the opposite steps for `BECLR`, first removing the patchwise masking (denoted as `BECLR−`) and then also the symmetric loss (denoted as `BECLR−−`), noticing that both slightly degrade the performance, as seen in Table 14, which confirms our design choices to include them. Similarly, we also remove `OpTA` from `BECLR`'s inference stage. The most important takeaway here is that the most degraded version of `BECLR` (`BECLR−− w/o OpTA`) still outperforms the best enhanced version of PsCo ( `PsCo++ w/ OpTA`), which we believe offers an additional perspective on the advantages of adopting `BECLR` over PsCo.

## F.3 COMPUTATIONAL COMPLEXITY

Finally, we also compare `BECLR` and PsCo in terms of model size (total number of parameters in M) and inference times (in sec/episode), when using the same ResNet-18 backbone architecture. The results are summarized in Table 15. We notice that `BECLR`'s total parameter count is, in fact, lower than that of PsCo. Regarding inference time, `BECLR` is slightly slower in comparison, but this difference is negligible in real-time inference scenarios.

Table 15: Comparison of the computational complexity of `BECLR` with PsCo (Jang et al., 2022) in terms of total model parameter count and inference times.

| Method | Backbone | Model Total Parameter Count (M) | Inference Time (sec/episode) |
|---|---|---|---|
| PsCo (Jang et al., 2022) | RN18 | 30.766 | 0.082 |
| BECLR (Ours) | RN18 | 24.199 | 0.216 |

# G    COMPARISON WITH SUPERVISED FSL

In this work we have focused on unsupervised few-shot learning and demonstrated how `BECLR` sets a new state-of-the-art in this exciting space. In this section, as an additional study, we are also comparing `BECLR` with a group of most recent supervised FSL baselines, which also have access to the base-class labels in the pretraining stage.

## G.1    PERFORMANCE EVALUATION

In Table 16 we compare `BECLR` with seven recent baselines from the supervised FSL state-of-the-art (Lee et al., 2019; Bateni et al., 2022; He et al., 2022; Hiller et al., 2022; Bendou et al., 2022; Singh & Jamali-Rad, 2022; Hu et al., 2023b), in terms of their in-domain performance on miniImageNet and tieredImageNet. As can be seen, some of these supervised methods do outperform `BECLR`, yet these baselines are heavily engineered towards the target dataset in the in-domain setting (where the target classes still originate from the pretraining dataset). Another interesting observation here is that it turns out the top performing supervised baselines are all *transductive* methodologies (i.e., learn from both labeled and unlabeled data at the same time), and such a transductive episodic pretraining cannot be established in a fully unsupervised pretraining strategy as in `BECLR` Notice that `BECLR` can even outperform recent inductive supervised FSL approaches, *even without access to base-class labels* during pretraining.

Table 16: Accuracies (in % ± std) on miniImageNet and tieredImageNet compared against supervised FSL baselines. Pretrainig Setting: unsupervised (`Unsup.`) and supervised (`Sup.`) pretraining. Approach: `Ind.`: inductive setting, `Transd.`: transductive setting. Style: **best** and second best.

| | | | miniImageNet | | tieredImageNet | |
|---|---|---|---|---|---|---|
| **Method** | **Setting** | **Approach** | **5 way 1 shot** | **5 way 5 shot** | **5 way 1 shot** | **5 way 5 shot** |
| **BECLR (Ours)** | `Unsup.` | `Ind.` | **80.57** ± 0.57 | **87.82** ± 0.29 | **81.69** ± 0.61 | 87.86 ± 0.32 |
| MetaOptNet (Lee et al., 2019) | `Sup.` | `Ind.` | 64.09 ± 0.62 | 80.00 ± 0.45 | 65.99 ± 0.72 | 81.56 ± 0.53 |
| HCTransformers (He et al., 2022) | `Sup.` | `Ind.` | 74.74 ± 0.17 | 85.66 ± 0.10 | 79.67 ± 0.20 | 89.27 ± 0.13 |
| FewTURE (Hiller et al., 2022) | `Sup.` | `Ind.` | 72.40 ± 0.78 | 86.38 ± 0.49 | 76.32 ± 0.87 | **89.96** ± 0.55 |
| EASY (inductive) (Bendou et al., 2022) | `Sup.` | `Ind.` | 70.63 ± 0.20 | 86.28 ± 0.12 | 74.31 ± 0.22 | 87.86 ± 0.15 |
| EASY (transductive) (Bendou et al., 2022) | `Sup.` | `Transd.` | 82.31 ± 0.24 | 88.57 ± 0.12 | 83.98 ± 0.24 | 89.26 ± 0.14 |
| Transductive CNAPS (Bateni et al., 2022) | `Sup.` | `Transd.` | 55.60 ± 0.90 | 73.10 ± 0.70 | 65.90 ± 1.10 | 81.80 ± 0.70 |
| BAVARDAGE (Hu et al., 2023b) | `Sup.` | `Transd.` | 84.80 ± 0.25 | 91.65 ± 0.10 | 85.20 ± 0.25 | 90.41 ± 0.14 |
| TRIDENT (Singh & Jamali-Rad, 2022) | `Sup.` | `Transd.` | **86.11** ± 0.59 | **95.95** ± 0.28 | **86.97** ± 0.50 | **96.57** ± 0.17 |

## G.2    ADDITIONAL INTUITION

Let us now try to provide some high-level intuition as to why `BECLR` can outperform such supervised baselines. We argue that self-supervised pretraining helps generalization to the unseen classes, whereas supervised training heavily tailors the model towards the pretraining classes. This is also evidenced by their optimization objectives. In particular, supervised pretraining maximizes the mutual information $I(Z, y)$ between representations $Z$ and base-class labels $y$, whereas `BECLR` maximizes the mutual information $I(Z_1, Z_2)$ between different augmented views $Z_1, Z_2$ of the input images $X$, which is a lower bound of the mutual information $I(Z, X)$ between representations $Z$ and raw data/ images $X$. As such, `BECLR` *potentially* has a higher capacity to learn more discriminative and generalizable features. By the way, this is not the case only for `BECLR`, many other unsupervised FSL approaches report similar behavior, e.g. (Chen et al., 2021a; Lu et al., 2022; Hu et al., 2023a). Next to that, pure self-supervised learning approaches also report a similar observation where they can outperform supervised counterparts due to better generalization to different downstream tasks, e.g. (Hénaff et al., 2021; Zhang et al., 2022).

Another notable reason why `BECLR` can outperform some of its supervised counterparts is our novel module `OpTA` specifically designed to address sample bias, a problem that both supervised and unsupervised FSL approaches suffer from and typically overlook. So comes our claim that the proposed `OpTA` should becomes an integral part of all (U-)FSL approaches, especially in low-shot scenarios where FSL approaches suffer from samples bias the most.

