# OpenReview forum: "BECLR: Batch Enhanced Contrastive Few-Shot Learning"
_ICLR.cc/2024/Conference — ICLR 2024 spotlight_

### Official Review · Reviewer_wMvt · 2023-10-30

**Soundness:** 3 good
**Presentation:** 3 good
**Contribution:** 3 good
**Rating:** 6
**Confidence:** 4

**Summary:**

The paper proposes a framework that comprises several part to better extract representations from images under the assumption that one single image in a batch corresponds to just one unique class. And also to address the issue of sample bias when few examples are playing crucial role even in cases where the sample is way higher.
\
The main contributions are: \

a) the end-to-end framework terms BECLR that incorporates the modules that address sample bias and expands single image instance discrimination \

b) Very good results in a number of benchmarks.

**Strengths:**

The paper is detailed and relatively easy to follow. The underlying method is well explained and motivated and is substantiated by several experiments and ablations. \

The results are promising and impressive and the method is well grounded.

**Weaknesses:**

The evaluation should have considered the various components in separation and with respect to other methods.
E.g. is OPTA that improves performance or perhaps adding another base line method might have had a similar performance. \
I think in such multi-component frameworks this is an issue usually. \

**Questions:**

a) are the results presented throughout the paper are primarily based on the end-to-end process that involves all components, including OPTA? \
b) To be fair we need to see how other baselines work when replacing OPTA, e.g. other methods and/or linear evaluation principles for FSL. OPTA seems to have a considerable effect but is that down to the method itself or other baseline methods added to this framework could have a similar effect? \
c) Is there a reason why you did not consider running on the entire imagenet?

Typos: \
there are several typos throughout the paper - e.g. abstract "Critical" not "Clinical", "downstream" not "downstrea" and others.

---

> ### Author Response · Authors · 2023-11-16
> **Response to Reviewer wMvt (1/3)**
>
> We thank the reviewer for their elaborate and constructive feedback. We are pleased to note that the reviewer finds the key proposed ideas grounded, the motivations behind $\texttt{BECLR}$ and its narrative well explained, and our extensive experimentation and ablation studies impressive and promising. We recognize the reviewer's suggestions around further experimentation by adding our novel proposed module (called $\texttt{OpTA}$) on top of other baselines, which nicely aligns with our broader impact statements, to substantiate the significance of its impact. Intrigued by these insightful suggestions, we have run new experimentation and address the reviewer's remarks in a point-to-point fashion in the following, as well as in the (updated) revised draft in $\textcolor{blue}{blue}$.
>
> ---
>
> **[W]**: The evaluation should have considered various components in separation and with respect to other methods. Adding $\texttt{OpTA}$ on other baselines. \
> **[AW]**: Thank you for this insightful comment, and indeed you are right, this is usually what happens in multi-component approaches. Let us first explain a few points and address your concerns.
>
> - First of all, please note that $\texttt{OpTA}$ **is not an existing module we adopt, but a key component** (see Key Idea II, in Section 1) and part of the novelty in the end-to-end U-FSL pipeline of $\texttt{BECLR}$. Even though in our broader impact and concluding remarks we actually propose $\texttt{OpTA}$ becoming an essential part of any FSL approach (regardless of being supervised or unsupervised), adding it on top of other prior art, cannot then be considered the same baseline anymore but **an enhanced version of it using part of our proposition**.
>
> - Related to this, we already have an **ablation study exactly dedicated to your point regarding component-wise analysis**. Please have a look at  Table $5$ in Section 5.2 (Ablation Studies) in the main text, where we study the robustness of $\texttt{BECLR}$ by sequentially adding essential components (including $\texttt{OpTA}$) and how this affects our end-to-end performance. This already allows us to compare against other baselines in term of performance.
>
> - Next to that, we also already take the opposite route in Fig. 5 in Section 5.1 (Evaluation Results) and **demonstrate that removing $\texttt{OpTA}$ from $\texttt{BECLR}$ still gives us state-of-the-art performance** compared to prior art. In this setting, both $\texttt{BECLR}$ and competing baselines **follow identical training and testing settings**, without applying $\texttt{OpTA}$ in the inference stage, thus further ensuring fairness in our comparisons.
>
> - We will build on this direction in the following **[AQb]** to thoroughly address your interesting point.
>
> ---
>
> **[Qa]**: Are the results, presented throughout the paper, based on the end-to-end process including $\texttt{OpTA}$?
>
> **[Aa]**: Yes they are, as $\texttt{BECLR}$ is an end-to-end U-FSL approach that encompasses both $\texttt{DyCE}$ and $\texttt{OpTA}$ as its key components. It should be mentioned, that this is the case for all competing baselines out there, since the U-FSL problem is solved in two sequential stages: pretraining and inference (or testing if you will), as discussed in Section 3 (Problem Statement: U-FSL). Therefore **any reported results for $\texttt{BECLR}$ and other baselines are based on the end-to-end process**. However, as we discussed earlier, to further study the robustness and versatility of $\texttt{BECLR}$, we also present its downstream performance without some of its key components, including $\texttt{OpTA}$ (see Table $5$ and Fig. $5$), also compared against the end-to-end performance of other baselines. More on this in the following.

---

> > ### Comment · Reviewer_wMvt · 2023-11-22
> > **Please elaborate**
> >
> > Can you please elaborate on the second bullet point above?
> > My point was exactly this; Table 5 in section 5.2 (by the way Table 6 appears first in the text) shows at the very last step how adding OPTA affects performance....one can interpret that as adding 13% of extra performance on 1-shot...however that cannot be a baseline comparison as this result merely shows what the benefit of OPTA is but not how a difference method in lieu of OPTA as a fourth stage in the process would compare.
> >
> > Say you replace OPTA, and performance still goes up at the same rate then what's the point of having OPTA. Is this ablation included somewhere? To me this type of ablation would involve freezing the first three steps, i.e. Masking, EMA teacher, and DyCE, and then testing various components alongside OPTA

---

> > > ### Comment · Reviewer_wMvt · 2023-11-22
> > > **increase score**
> > >
> > > Having said the above, I am happy with the rest of the responses so will increase my score....but table 5 still remains unclear to me.

---

> > > ### Author Response · Authors · 2023-11-22
> > > **Response to Reviewer wMvt - Round 2 (1/1)**
> > >
> > > We sincerely appreciate your insightful comment and that you have increased your score. We are determined to ensure Table 5 is fully clear, and we are happy to take any further actions you might still suggest. To do so, let us further elaborate on your remarks:
> > >
> > > ---
> > >
> > > - Regarding **ablations with a different method in lieu of $\texttt{OpTA}$**, as mentioned in Sections 1 (Introduction) and 2 (Related Work) the problem of sample bias remains mostly overlooked by U-FSL baselines. This means that, to the best of our knowledge, **there is no U-FSL baseline that introduces a module, similar to $\texttt{OpTA}$, to directly target the sample bias in the inference stage of U-FSL**. This is the very reason why we do not include an ablation where we would replace our proposed $\texttt{OpTA}$ with alternative pre-existing inference modules. In contrast, in the inference stage most U-FSL baselines either directly fit a logistic classifier on top of the pretrained feature encoder e.g. [1,2] or learn a linear classification layer via fine-tuning e.g. [3,4]. That being said, the reported results for all baselines and $\texttt{BECLR}$ are based on their end-to-end performance, i.e. after both the pretraining (for learning the feature encoder) and inference stage (for predicting query images on test episodes). Finally, note that $\texttt{OpTA}$ does not introduce any additional trainable parameters nor any fine-tuning, but relies on a deterministic transportation to align the distributions of support and query sets.
> > >
> > > - Indeed as you rightfully pointed out, Table 5 is an ablation study and does not concentrate on benchmarking against other baselines but delving deeper into the robustness of $\texttt{BECLR}$ and the impact of including its individual components. As such, **if we were to remove $\texttt{OpTA}$, our end-to-end performance** would be the one reported on the $5$th row of Table 5 and **would not not still go up**, since $\texttt{BECLR}$'s predictions would be prone to sample bias, as explained in Sections 1 (Introduction) and 2 (Related Work). To further clarify, rows $5$ and $6$ in Table 5 utilize the same pretrained model (which included both masking, EMA and $\texttt{DyCE}$ in the pretraining stage). Then, in the inference stage for the model without $\texttt{OpTA}$ and a given test episode, we extract query and support embeddings (using the pretrained feature encoder) and use the support set prototypes (class averages) to simply fit a logistic classifier. Instead, when $\texttt{OpTA}$ is also involved, it is utilized to transport the support set in the query set's domain, before using the modified support set prototypes for fitting the classifier. That being said, **$\texttt{BECLR}$ even without $\texttt{OpTA}$ can still be compared with and outperform the rest of U-FSL baselines** (see Fig. 5 and the Table above), since their training and evaluation settings remain identical and facilitate a fair comparison.
> > >
> > > - Regarding the **benefit of having $\texttt{OpTA}$**, as you mentioned, we demonstrate in Table 5 how it **offers a significant performance boost for $\texttt{BECLR}$**. In fact, in the table above in **[Ab]** (and Table 4 in the revised draft), we also confirm that **$\texttt{OpTA}$ is agnostic to the choice of U-FSL pretraining method and can offer meaningful end-to-end performance gains for different U-FSL baselines**. We argue this corroborates our claim that $\texttt{OpTA}$ is a **novel add-on module** that should be an integral part (in the inference stage) of all (U-)FSL approaches, especially in the challenging lower-shot scenarios, e.g. $1$-shot. We believe that the absence of alternative (inference) modules in the U-FSL literature similar to the proposed $\texttt{OpTA}$  further highlights its impact and novelty.
> > >
> > > ---
> > >
> > > Do these elaborations further clarify (i) the reasoning behind Table 5 and (ii) that there are no alternative modules similar to $\texttt{OpTA}$ to ablate on? Please let us know if this addresses your question.
> > >
> > > ---
> > >
> > > [1] Lu et al., Self-supervision can be a good few-shot learner, ECCV 2022 \
> > > [2] Wang et al., Contrastive prototypical network with Wasserstein confidence penalty, ECCV 2022 \
> > > [3] Chen et al., Few-Shot Learning with Part Discovery and Augmentation from Unlabeled Images, IJCAI 2021\
> > > [4] Jang et.al., Unsupervised Meta-learning via Few-shot Pseudo-supervised Contrastive Learning, ICLR 2023

---

> ### Author Response · Authors · 2023-11-16
> **Response to Reviewer wMvt (2/3)**
>
> **[Qb]**: To be fair we need to see how other baselines work when replacing $\texttt{OpTA}$. Adding $\texttt{OpTA}$ on other baselines, still an effective approach? \
> **[Ab]**: This is a great suggestion.
> - With all due respect, however, we do not think adding our innovative module on other baselines helps fairness in comparisons, but we totally agree that this substantiates the effectiveness of $\texttt{OpTA}$ and solidifies our claim on its broader impact. To address your insightful suggestion, we have put tremendous effort to conduct additional experiments, during this rebuttal period.
> - We picked a suite of recent prior art in U-FSL [1-4], along with two established SSL baselines [5-6] and reproduced their results, comparing to $\texttt{BECLR}$ without $\texttt{OpTA}$, as well as per your suggestion **produced new results by plugging in $\texttt{OpTA}$** in the inference stage **for all these competing baselines**. The results (see table below) demonstrate three important points: (i) **$\texttt{BECLR w/o OpTA}$ still outperforms all those prior art**, and (ii) **$\texttt{OpTA}$ is in fact agnostic to the choice of pretraining method**, since adding it on top of prior art has considerable impact on their enhanced methodology. We argue that this is a significant point in favor of our claim that $\texttt{OpTA}$ should become an integral part of all (U-)FSL approaches. (iii) There still exists a margin between enhanced prior art and $\texttt{BECLR}$, corroborating that **it is not just $\texttt{OpTA}$ that has a meaningful effect but also $\texttt{DyCE}$ and our pretraining methodology**. Notably, in Fig.7 in the main text we already demonstrate that $\texttt{DyCE}$ and $\texttt{OpTA}$ are heavily inter-dependent and significantly affect each other's impact. To address your concern, we have included these additional empirical results and discussion in the main text in Section 5 (Evaluation Results) of the revised manuscript.
>
> \begin{array}{l|c|cc|cc}
> \hline&& \text{miniImageNet}& \text{miniImageNet} & \text{tieredImageNet}& \text{tieredImageNet}\newline \hline
> \text{Method}& \text{Backbone}& \text{5 way 1 shot}& \text{5 way 5 shot}& \text{5 way 1 shot}& \text{5 way 5 shot}\newline \hline
> \texttt{NNCLR}& \text{ResNet-18} & 63.33 ± 0.53& 80.75 ± 0.25& 65.46 ± 0.55 & 81.40 ± 0.22 \newline
> \texttt{SwAV}& \text{ResNet-18} & 59.84 ± 0.52& 78.23 ± 0.26& 65.26 ± 0.53 & 81.73 ± 0.24 \newline
> \texttt{CPNWCP}& \text{ResNet-18}& 53.81 ± 0.65& 69.61 ± 0.56 & 47.63 ± 0.19 & 64.18 ± 0.17\newline
> \texttt{HMS}& \text{ResNet-18} 	& 58.20 ± 0.23& 75.77 ± 0.16 & 58.42 ± 0.25& 75.85 ± 0.18\newline
> \texttt{PsCo}& \text{ResNet-18}& 47.24 ± 0.56& 65.48 ± 0.28 & 54.33 ± 0.54& 69.73 ± 0.49\newline
> \texttt{UniSiam}& \text{ResNet-18}& 63.26 ± 0.36& 81.13 ± 0.26 & 65.18 ± 0.39 & 82.28 ± 0.29\newline
> \texttt{BECLR w/o OpTA} & \text{ResNet-18} & \bf{66.14 ± 0.43}& \bf{84.32 ± 0.27} & \bf{67.86 ± 0.50} & \bf{84.16 ± 0.36}\newline \hline
> \texttt{NNCLR+OpTA} & \text{ResNet-18} & 72.62 ± 0.64 & 82.19 ± 0.35& 73.16 ± 0.62& 82.82 ± 0.33 \newline
> \texttt{SwAV+OpTA}& \text{ResNet-18} & 70.36 ± 0.61 & 81.68 ± 0.36 & 73.60 ± 0.63& 82.37 ± 0.34 \newline
> \texttt{CPNWCP+OpTA}& \text{ResNet-18} & 60.45 ± 0.81 & 75.84 ± 0.56 & 55.05 ± 0.31& 72.91 ± 0.26 \newline
> \texttt{HMS+OpTA}& \text{ResNet-18} & 69.85 ± 0.42 & 80.77 ± 0.35& 71.75 ± 0.43& 81.32 ± 0.34\newline
> \texttt{PsCo+OpTA}& \text{ResNet-18} & 52.89 ± 0.71 & 67.42 ± 0.54& 57.46 ± 0.59& 70.70 ± 0.45 \newline
> \texttt{UniSiam+OpTA}& \text{ResNet-18} & 72.54 ± 0.61 & 82.46 ± 0.32 & 73.37 ± 0.64& 82.64 ± 0.64 \newline
> \texttt{BECLR}& \text{ResNet-18}& \bf{75.74 ± 0.62} & \bf{84.93 ± 0.33}& \bf{76.44 ± 0.66} & \bf{84.85 ± 0.37}\newline \hline
> \end{array}
>
> [1] Lu et al., Self-supervision can be a good few-shot learner, ECCV 2022 \
> [2] Wang et al., Contrastive prototypical network with Wasserstein confidence penalty, ECCV 2022 \
> [3] Li et al., Unsupervised few-shot image classification by learning features into clustering space,  ECCV 2022 \
> [4] Jang et.al., Unsupervised Meta-learning via Few-shot Pseudo-supervised Contrastive Learning, ICLR 2023 \
> [5] Dwibedi et al., With a little help from my friends: Nearest-neighbor contrastive learning of visual representations, ICCV 2021 \
> [6] Caron et al., Unsupervised learning of visual features by contrasting cluster assignments, NeurIPS 2020

---

> ### Author Response · Authors · 2023-11-16
> **Response to Reviewer wMvt (3/3)**
>
> **[Qc]**: Running experiments on the entire ImageNet? \
> **[Ac]**: This is a great question!
> - To the best of our knowledge, no existing U-FSL baseline reports results on full ImageNet. FSL benchmarks define specific FSL-oriented samplers and settings which are tailored towards miniImageNet, tieredImageNet, CIFAR-FS, and FC-100 (in-domain) and miniImagenet → CDFSL and miniImagenet → CUB (cross-domain). These are **THE benchmarks adopted by $99.9$**% **of the U-FSL approaches** out there. Notably, in a separate study dedicated to SSL (and not U-FSL anymore) we are exploring the capacity of $\texttt{BECLR}$ on the entire ImageNet, but that is outside the scope of this paper. If you strongly advise, we can include some of those preliminary results in the Appendix.
>
> ---
> **[C]**: There are several typos throughout the paper. \
> **[AC]**: Thank you for these astute observations. Regarding the word "clinical" in the Abstract we meant that the problem of sample bias is inherent to (U-)FSL, yet we agree that critical would be a better choice of word. Regarding the typo in "downstream", this is indeed a typo that has happened on the last revision of the paper (it is not present in previous versions). Following your comment, we have carefully reviewed the revised manuscript multiple times to ensure the absence of typos.

---

> > ### Author Response · Authors · 2023-11-20
> > **Gentle Follow Up on Our Responses**
> >
> > We wanted to kindly make a follow up on the elaborate response and revisions we have provided and see if they address your concerns. We would be more than happy to provide further clarifications and revisions if you have any more questions or concerns, and if not we would greatly appreciate it if you would please re-evaluate our paper's score. Thank you again for your reviews which helped us tremendously to improve our paper!

---

### Official Review · Reviewer_WH4q · 2023-10-30

**Soundness:** 3 good
**Presentation:** 3 good
**Contribution:** 2 fair
**Rating:** 6
**Confidence:** 5

**Summary:**

In the presented paper, the authors identify two primary limitations in the existing Unsupervised Few-Shot Learning (U-FSL) methods and introduce an integrated solution named BECLR. BECLR incorporates two novel components: a dynamic clustered memory module named DyCE, designed to improve positive sampling in contrastive learning, and an efficient distribution alignment strategy termed OpTA, devised to counteract sample bias in U-FSL. While BECLR is tailored for U-FSL, the authors highlight DyCE's potential broader applications in general self-supervised learning, noting its superior performance even without OpTA, and advocate for the inclusion of OpTA in all U-FSL methods, particularly in low-shot scenarios.

**Strengths:**

**Originality**: The paper showcases a distinct approach by addressing recognized limitations in U-FSL and introducing the BECLR solution, merging existing concepts innovatively and presenting fresh modules like DyCE and OpTA.

**Quality**: The research is robust, with DyCE and OpTA being methodically developed and their effectiveness demonstrated through comparisons with established methods like SwaV, SimSiam, and NNCLR.

**Clarity**: The authors articulate their findings and methodologies clearly, ensuring that readers can grasp the intricacies of BECLR, DyCE, and OpTA without ambiguity.

**Significance**: With its potential to redefine self-supervised learning and its implications for U-FSL, especially in few-shot scenarios, the paper holds substantial importance in advancing the field and offers a direction for future research.

**Weaknesses:**

1. The overarching idea and structure of BECLR bear a striking resemblance to PsCo, particularly when observing Figure 2. It appears that the primary distinction is the concatenation of different views of 'X' from PsCo and the addition of a dynamic clustered memory. To differentiate their work more effectively, the authors should provide a comprehensive comparison with PsCo in both the introduction and related work sections, highlighting the unique aspects of their approach.

2. The notation used in the algorithmic section needs elucidation. Clearly defining each symbol would make it more accessible and allow readers to follow the content with greater ease.

3. The experimental section could benefit from an additional test: an evaluation of performance without merging 'X'. It would also be insightful to see results when PsCo is merged and when masking is applied, offering a more comprehensive understanding of the method's robustness and versatility.

**Questions:**

1. **Model Parameter Updates during Testing:** During the testing phase, are there any updates required for the model's parameters? If yes, how many times is the model updated?

2. **Comparison with PsCo in Terms of Model Size and Inference Time:** How does BECLR compare to PsCo regarding the number of parameters in the model and the inference time? A direct comparison would provide clarity on the efficiency and scalability of BECLR.

3. **Data Utilized for MiniImagenet Pretraining:** When pretraining with miniImagenet, which specific datasets were employed? Was the entire miniImagenet dataset utilized for this purpose?

Suggestions:

- **Enhanced Clarity on Parameter Update Mechanism:** A deeper dive into the model's updating mechanism during testing would be valuable. This would provide insights into the adaptability and robustness of the model, especially in real-world scenarios where data dynamics may vary.

- **Detailed Comparative Analysis with PsCo:** Given the similarities noted between BECLR and PsCo, a side-by-side comparison in terms of model parameters and inference time would offer readers a clearer perspective on the advantages and potential trade-offs of adopting BECLR.

---

> ### Author Response · Authors · 2023-11-16
> **Response to Reviewer WH4q (1/3)**
>
> We truly appreciate the reviewer's elaborate and constructive feedback. We are delighted to note that the reviewer accentuates on the novelty of the key ideas, clarity and coherence of the narrative, effectiveness of $\texttt{BECLR}$ demonstrated through extensive experimentation (defining it as SoTA across all existing FSL benchmarks), as well as on the significance of the work potentially "redefining self supervised learning'' and U-FSL. We recognize the reviewer's constructive suggestions around more elaborate comparison with PsCo (in terms of performance, complexity, and inference time) as well as further clarity on parameter update mechanism. To address these remarks, we have put tremendous effort in running new experimentation and providing a point-to-point response, which in turn is reflected in the (updated) revised draft in $\textcolor{blue}{blue}$.
>
> ---
> **[W1-a]**: Resemblance to PsCo, unique aspects of $\texttt{BECLR}$, and providing comparison with PsCo in the introduction and related work. \
> **[AW1-a]**: Thank you for this helpful pointer.
> - First of all, we dove deeper into PsCo and set up its pipeline for reproducing its reported results and conducting additional experiments. $\texttt{BECLR}$ and PsCo indeed share some similarities, in that both methods utilize a student-teacher momentum architecture, a memory module of past representations, some form of contrastive loss, and optimal transport (even though for different purposes). Note that none of these components are unique to neither PsCo nor $\texttt{BECLR}$ and can be found in the overall SSL literature [1-3].
> - Let us now expand on their discrepancies and **unique aspects of $\texttt{BECLR}$**: (i) PsCo is based on meta learning, constructing few-shot classification (i.e., $N$-way $K$-shot) tasks, during meta-training and relying on fine-tuning for rapidly adapting to novel tasks during meta-testing. In contrast, $\texttt{BECLR}$ is a **contrastive framework based on metric/transfer learning**, which focuses on representation quality and relies on $\texttt{OpTA}$ for transferring to novel tasks (no few-shot tasks during pretraining, no fine-tuning during inference). (ii) PsCo utilizes a simple FIFO memory queue (oblivious to class-level information), while $\texttt{BECLR}$ maintains a **clustered highly-separable** (see Figure 6) **memory module** in $\texttt{DyCE}$, which (after an adaptation period) is used for sampling meaningful positives. (iii) PsCo applies optimal transport for creating pseudo-labels (directly from the unstructured memory) for $N*K$ support embeddings, in turn used as a supervisory signal to enforce consistency between support (drawn from teacher) and query (student) embeddings. In contrast, $\texttt{BECLR}$ artificially enhances the batch with additional positives and applies an instance-level contrastive loss to enforce consistency between the original and enhanced positive pairs. After each training iteration, optimal transport is applied to **update the stored clusters within $\texttt{DyCE}$ in an equipartitioned fashion** with embeddings from the current batch. (iv) Finally, $\texttt{BECLR}$ **also incorporates optimal transport (in $\texttt{OpTA}$) for aligning the distributions between support and query sets**, during inference, which shares no similarity with the pipeline of PsCo. Following your suggestion, we added further discussion on the comparison with PsCo both in Sections 2 (Related Work) and 4.1 (Unsupervised Pretraining) of the revised draft.
>
> [1] Wang et al., Contrastive prototypical network with Wasserstein confidence penalty, ECCV 2022 \
> [2] Grill et al. Bootstrap your own latent-a new approach to self-supervised learning, NeurIPS 2020 \
> [3] Dwibedi et al. With a little help from my friends: Nearest-neighbor contrastive learning of visual representations, ICCV 2021
>
> ---
> **[W1-b]**: It appears that the primary distinction is the concatenation of different views of $X$ from PsCo and the addition of a dynamic clustered memory. \
> **[AW1-b]**: Great remark. Let us expand further on this point:
> - The reasoning behind the concatenation of different views of $X$, is to pass both views to both student/teacher networks to accommodate our symmetric contrastive loss (i.e., infoNCE loss between $Z_1^{S}$-$Z_2^{T}$ and $Z_2^{S}$-$Z_1^{T}$). The symmetricity in the loss has been shown to yield a minor performance boost [1], but does not constitute the primary distinction between PsCo and $\texttt{BECLR}$ as explained in **[AW1-a]**. The effect of adding a symmetric loss to PsCo is also discussed in greater detail in the following **[AW3]**.
> - **The main novelty** and distinction lies indeed in our **dynamic clustered memory** ($\texttt{DyCE}$) and our **distribution alignment module** ($\texttt{OpTA}$), which are the two Key Ideas of our work, both contributing to the overall U-FSL performance of $\texttt{BECLR}$ (see Table 5).
>
> [1] Chen et. al. Exploring simple siamese representation learning, CVPR 2021

---

> ### Author Response · Authors · 2023-11-16
> **Response to Reviewer WH4q (2/3)**
>
> **[W2]**: Elucidation of the notation in the Algorithm. \
> **[AW2]**: Thank you for the constructive recommendation.
>  - We did our best to define every variable in the main text and also included a step-by-step walk-through for Algorithm 2. However, we agree this could have been done better. Per your suggestion, we (i) **now include a thorough walk-through of Algorithm 1**, (ii) have made **modifications in both Algorithm 2 and its explanation** to provide more elucidation on the inner mechanics of $\texttt{DyCE}$, and (iii) made sure that **every single variable and symbol (for both algorithms) are properly defined** in the main text. These revisions are included in Section 4.1 (Unsupervised Pretraining) of the updated manuscript.
>
> - Please also have a look at Algorithms 3 and 4, already existing in Appendix E, where we also provide PyTorch-like pseudocodes for Algorithms 1 and 2 for further elucidation.
>
> ---
> **[W3]**: Additional test by including/ exlcuding masking and concatenating augmented views of the input $X$.
>
> **[AW3]**: This is a fabulous remark and intrigued us to run new experimentation to fully address your concern here.
>
> - Before we dive in, note that PsCo reports performance with a Conv5 backbone, even though in exploring the codebase we understood it can readily accept ResNet-18, as well. So, in the below table we present an interesting experiment. We first mark the original reported performance of PsCo with a Conv5 backbone as well as its reproduction with ResNet-18. We then modify the **loss for PsCo to be symmetric** (allowing both augmentations of $X$ to pass through both branches) similar to $\texttt{BECLR}$, (this is denoted as $\texttt{PsCo+}$). Next, we **also add patch-wise masking** to PsCo (denoted as $\texttt{PsCo++}$) and as an additional step we have **also added our novel module $\texttt{OpTA}$** on top of all these models ($\texttt{PsCo w/ OpTA}$), during inference. We observe that neither the symmetric loss nor masking are offering meaningful improvements in the performance of PsCo. On the contrary $\texttt{OpTA}$ yields a significant performance boost (up to $6.5$% in the $1$-shot setting that sample bias is most severe). This corroborates our claim that our proposed $\texttt{OpTA}$ should be considered as an add-on module to every FSL approach out there (Key Idea II).
>
> - **As an additional robustness study** we take the opposite steps for $\texttt{BECLR}$, first **removing the patch-wise masking** (denoted as $\texttt{BECLR-}$) and then **also the symmetric loss** (denoted as $\texttt{BECLR--}$), noticing that both degrade the performance slightly (see table below), which confirms our design choices to include them. Similarly, we also remove $\texttt{OpTA}$ from $\texttt{BECLR}$'s inference stage. The most important takeaway here: **the most degraded version of $\texttt{BECLR}$** ($\texttt{BECLR-- w/o OpTA}$) **still outperforms the best enhanced version of PsCo**( $\texttt{PsCo++ w/ OpTA}$), even without $\texttt{OpTA}$, which we think offers additional perspective on the advantages of adopting $\texttt{BECLR}$ over PsCo.
>
> - To reiterate, we believe $\texttt{BECLR}$ defines the new SoTA over ALL U-FSL baselines across all existing benchmarks dedicated to FSL (to the best of our knowledge). We have added this table and discussion in Appendix F (In-Depth Comparison with PsCo), as this study really focuses on a detailed comparison with a single baseline, but we are happy to pull this up into the main text of the revised version, in case you strongly advise to do so. We do hope this addresses your concern.
> \begin{array}{l|c|cc}
> \hline
> \text{Method}& \text{Backbone}& \text{5 way 1 shot} & \text{5 way 5 shot} \newline \hline
> \texttt{PsCo}& \text{Conv5}& 46.70 ± 0.42& 63.26 ± 0.37 \newline
> \texttt{PsCo}& \text{ResNet-18}& 47.24 ± 0.46& 65.48 ± 0.38\newline
> \texttt{PsCo+}& \text{ResNet-18}& 47.86 ± 0.44& 65.95 ± 0.37\newline
> \texttt{PsCo++}& \text{ResNet-18}& 47.58 ± 0.45& 65.74 ± 0.38\newline \hline
> \texttt{PsCo w/ OpTA}& \text{ResNet-18} & 52.89 ± 0.61& 67.42 ± 0.51\newline
> \texttt{PsCo+ w/ OpTA}& \text{ResNet-18}& 54.43 ± 0.59& 68.31 ± 0.52\newline
> \texttt{PsCo++ w/ OpTA}& \text{ResNet-18}& 54.35 ± 0.60& 68.43 ± 0.52\newline \hline
> \texttt{BECLR}& \text{ResNet-18}& \bf{75.74 ± 0.62} & \bf{84.93 ± 0.33}\newline
> \texttt{BECLR-}& \text{ResNet-18} & 74.37 ± 0.61 & 84.19 ± 0.31\newline
> \texttt{BECLR--}& \text{ResNet-18} & 73.65 ± 0.61& 83.63 ± 0.31\newline \hline
> \texttt{BECLR w/o OpTA} & \text{ResNet-18}& 66.14 ± 0.43& 84.32 ± 0.27\newline
> \texttt{BECLR- w/o OpTA} & \text{ResNet-18}& 65.26 ± 0.41 & 83.68 ± 0.25\newline
> \texttt{BECLR-- w/o OpTA}& \text{ResNet-18} & 64.68 ± 0.44& 83.45 ± 0.26\newline \hline
> \end{array}

---

> > ### Comment · Reviewer_WH4q · 2023-11-18
> > **Thanks for your response.**
> >
> > I greatly appreciate your thorough response, which has addressed all my initial concerns. However, I still have one question. Is there a high-level intuition as to why your method significantly outperforms supervised few-shot learning? I understand the contributions and the implementation details of your paper, but I'm interested in any intuitive explanation you might have.
> >
> > Additionally, you mentioned comparisons with MetaOptNet and Transductive CNAPS. Could you possibly provide a comparison with some of the supervised few-shot learning methods that have emerged in the past year?
> >
> > Looking forward to your insights.

---

> > > ### Author Response · Authors · 2023-11-20
> > > **Response to Reviewer WH4q - Round 2 (1/2)**
> > >
> > > We also greatly appreciate the reviewers swift reaction and constructive conversation to improve the quality of the revised draft. We are delighted to hear that reviewer's **all initial concerns are already addressed**, and more than happy to provide further elaborations around the next questions in the following.
> > >
> > > ---
> > > **[Q4]**: Is there a high-level intuition as to why your method outperforms supervised FSL?\
> > > **[A4]**: This is indeed a great question and touches upon a core concept.
> > > - We believe the intuition behind why $\texttt{BECLR}$ outperforms the cited supervised baselines is that **self-supervised pretraining helps generalization to the unseen classes, whereas supervised training heavily tailors the model towards the pretraining classes.** This is also evidenced by their optimization objectives. In particular, supervised pretraining maximizes the mutual information $I(\bf{Z}, \bf{y})$ between representations $\bf{Z}$ and base-class labels $\bf{y}$, whereas $\texttt{BECLR}$ maximizes (a lower bound of) the mutual information $I(\bf{Z}, \bf{X})$ between representations $\bf{Z}$ and raw data/ images $\bf{X}$. As such, $\texttt{BECLR}$ _potentially_ has a higher capacity to learn more discriminative and generalizable features. By the way, this is not the case only for $\texttt{BECLR}$, many other unsupervised FSL approaches report similar behavior, e.g. [1,2]. Next to that, pure self-supervised learning approaches also report a similar observation where they can outperform supervised counterparts due to better generalization to different downstream tasks, e.g. [3,4].
> > > - Another notable reason why $\texttt{BECLR}$ can outperform some supervised counterparts is our novel module $\texttt{OpTA}$ specifically **designed to address sample bias, a problem that both supervised and unsupervised FSL approaches suffer from and typically overlook**. So comes our claim that the proposed $\texttt{OpTA}$ should becomes an integral part of all FSL approaches, especially in low-shot scenarios where FSL approaches suffer from samples bias the most.
> > >
> > > [1] Lu et al., Self-supervision can be a good few-shot learner, ECCV 2022\
> > > [2] Chen et al., Few-Shot Learning with Part Discovery and Augmentation from Unlabeled Images, IJCAI 2021\
> > > [3] Hénaff et al., Efficient visual pretraining with contrastive detection, ICCV 2021\
> > > [4] Zhang et al., DINO: DETR with Improved DeNoising Anchor Boxes for End-to-End Object Detection, ICLR 2022

---

> > > ### Author Response · Authors · 2023-11-20
> > > **Response to Reviewer WH4q - Round 2 (2/2)**
> > >
> > > **[Q5]**: Could you possibly provide a comparison with some of the supervised few-shot learning methods that have emerged in the past year?\
> > > **[A5]**: Thanks for this helpful remark.
> > > - The focus of our work is on U-FSL, and not supervised FSL, which is why only a limited pool of supervised baselines is provided in our comparisons, just for bench-marking purposes. That being said, following your suggestion, we have prepared the below table where we **compare $\texttt{BECLR}$ with a group of most recent** (from 2022-2023) **supervised baselines** [1-5], along with the pre-existing MetaOptNet [6] and Transductive CNAPS [7]. As can be seen, these methods indeed outperform $\texttt{BECLR}$, yet some of these methods are heavily engineered towards the target dataset in the in-domain setting (where the previously discussed in **[AQ4]** generalization gap is not that significant since the target classes still originate from the pretraining dataset). Another interesting observation here is that it turns out that the top performing supervised SoTA baselines are all _transductive_ methodologies (i.e. learn from both labeled and unlabeled data at the same time), and such a transductive episodic pretraining cannot be established in a fully unsupervised pretraining strategy as in $\texttt{BECLR}$. Notice that $\texttt{BECLR}$ **can even outperform recent inductive supervised FSL approaches, even without access to base-class labels during pretraining.**
> > > - The below table and discussion around it have been added in Table 16 in Appendix G (Comparison with Supervised FSL), since our main focus is about U-FSL and not supervised FSL. We do hope this fully addresses your insightful remarks.
> > >
> > > \begin{array}{lcccccc}
> > > \hline&&&\text{miniImageNet}&\text{miniImageNet}&\text{tieredImageNet}&\text{tieredImageNet}\newline\hline
> > > \text{Method}& \text{Setting}&\text{Approach}& \text{5 way 1 shot}& \text{5 way 5 shot}&\text{5 way 1 shot}&\text{5 way 5 shot}\newline \hline
> > > \texttt{BECLR}\text{ (Ours)}& \text{Unsup.}&\text{Inductive}&\bf{80.57 ± 0.57}&\bf{87.82 ± 0.29}&\bf{81.69 ± 0.61}&87.86 ± 0.32\newline
> > > \texttt{MetaOptNet}& \text{Sup.}&\text{Inductive}&64.09 ± 0.62&80.00 ± 0.45&65.99 ± 0.72&81.56 ± 0.53\newline
> > > \texttt{HCTransformers}&\text{Sup.}& \text{Inductive}&74.74 ± 0.17&85.66 ± 0.10&79.67 ± 0.20&89.27 ± 0.13\newline
> > > \texttt{FewTURE}&\text{Sup.}&\text{Inductive}&72.40 ± 0.78& 86.38 ± 0.49&76.32 ± 0.87&\bf{89.96 ± 0.55}\newline
> > > \texttt{EASY}\text{(inductive)}&\text{Sup.}&\text{Inductive}&70.63 ± 0.20&86.28 ± 0.12&74.31 ± 0.22&87.86 ± 0.15\newline \hline
> > > \texttt{EASY}\text{(transductive)}&\text{Sup.}&\text{Transductive}&82.31 ± 0.24&88.57 ± 0.12&83.98 ± 0.24&89.26 ± 0.14\newline
> > > \texttt{Transductive CNAPS}&\text{Sup.}&\text{Transductive}&55.60 ± 0.90&73.10 ± 0.70&65.90 ± 1.10&81.80 ± 0.70\newline
> > > \texttt{BAVARDAGE}&\text{Sup.}&\text{Transductive}&84.80 ± 0.25&91.65 ± 0.10&85.20 ± 0.25&90.41 ± 0.14\newline
> > > \texttt{TRIDENT}&\text{Sup.}&\text{Transductive}&\bf{86.11 ± 0.59}&\bf{95.95 ± 0.28}&\bf{86.97 ± 0.50}&\bf{96.57 ± 0.17}\newline\hline
> > > \end{array}
> > > [1] He et al., Attribute surrogates learning and spectral tokens pooling in transformers for few-shot learning, CVPR 2022\
> > > [2] Hiller et al., Rethinking generalization in few-shot classification, NeurIPS 2022\
> > > [3] Singh et al., Transductive Decoupled Variational Inference for Few-Shot Classification, TMLR 2022\
> > > [4] Bendou et al., Easy-ensemble augmented-shot-y-shaped learning: State-of-the-art few-shot classification with simple components, Journal of Imaging 2022\
> > > [5] Hu et al., Adaptive Dimension Reduction and Variational Inference for Transductive Few-Shot Classification, AISTATS 2023\
> > > [6] Lee et al., Meta-learning with differentiable convex optimization, CVPR 2019\
> > > [7] Bateni et al., Enhancing few-shot image classification with unlabelled examples, WACV 2022

---

> > > > ### Author Response · Authors · 2023-11-21
> > > > **Gentle Follow up on Our Responses**
> > > >
> > > > We wanted to kindly make a follow up on the elaborate response and revisions we have provided in the second round (and updated the paper twice according to your suggestions) and see if they address all your remaining questions/concerns. We would be more than happy to provide further clarifications and revisions if you have any more questions or concerns, and if not we would greatly appreciate it if you would please re-evaluate our paper's score. Thank you again for your reviews which helped us tremendously to improve our paper!

---

> > > > > ### Comment · Reviewer_WH4q · 2023-11-22
> > > > > **Thanks for your reply!**
> > > > >
> > > > > I am very satisfied with the author's response and the additional references provided, so I have increased my score to 6. I hope that in the new version, the author can include discussions of the experiments and literature.

---

> ### Author Response · Authors · 2023-11-16
> **Response to Reviewer WH4q (3/3)**
>
> **[Q1]**: Model parameter updates during test phase. \
> **[A1]**: Again, an important remark.
>
> - **We do not have any model updates nor any fine-tuning at test time.** Neither $\texttt{DyCE}$ nor $\texttt{OpTA}$ impose any extra learnable parameters, and that we believe is part of the beauty of $\texttt{BECLR}$. Following your suggestion, we have added a sentence to address this in Section 4.2 (Supervised Inference) of the revised draft.
>
> ---
>
> **[Q2**: Model size and inference time comparison against PsCo.  \
> **[A2]**: Yet another interesting recommendation.
>
> - Following your suggestion, during this rebuttal period **we have compared $\texttt{BECLR}$ and PsCo in terms of model size** (total number of parameters in M) **and inference times** (in sec/episode), when using the same ResNet-18 backbone architecture. The results are summarized in the table below. We notice that $\texttt{BECLR}$'s total parameter count is in fact lower than that of PsCo. Regarding inference time, $\texttt{BECLR}$ is slightly slower in comparison, yet this difference is negligible in real-time inference scenarios. This comparison has also been added in Appendix F (In-Depth Comparison with PsCo) of the revised draft, since it constitutes a comparison against a single U-FSL baseline. Please let us know if this sufficiently addresses your concern.
>
> - Please also have a look at Table 13 already existing in Appendix D (Complexity Analysis), where we compare $\texttt{BECLR}$ with different SSL contrastive approaches in terms of computational complexity (model size, training time, inference time)
>
>
> \begin{array}{lccc}
> \hline
> \text{Method} 		& \text{Backbone} 	& \text{Total Parameter Count (M)} 	& \text{Inference Time (sec/episode)} 	\newline \hline
> \texttt{PsCo}            	& \text{ResNet-18}     	& 30.766                        	& 0.082                               	\newline
> \texttt{BECLR}           	& \text{ResNet-18}     	& 24.199                        	& 0.216                               	\newline \hline
> \end{array}
>
> ---
>
> **[Q3]**: Data used for miniImageNet pretraining. \
> **[A3]**: Thank you for this helpful comment.
>
> - First of all, please have a look at Section B.1 (Dataset Details) already in Appendix B. Therein, we expand on the datasets that were used as part of the experimental process of $\texttt{BECLR}$, in terms of selected splits used for training, validation, and testing, number of classes and dataset size.
>
> - For miniImageNet in particular, out of its $100$ classes, we select $64$, $16$, and $20$ classes for training, validation, and testing, respectively, following the **predominantly adopted settings** [1-4]. The training, validation, and testing splits and classes are mutually exclusive, as is standard practice in (U-)FSL, to test the adaptation of the model to novel tasks (unseen during training). To explicitly address your question, **when pretraining $\texttt{BECLR}$ on miniImageNet, only the training split is employed and not the entire miniImageNet dataset, nor any additional datasets.**
>
> [1] Ravi et.al., Optimization as a model for few-shot learning, ICLR 2016 \
> [2] Jang et.al., Unsupervised Meta-learning via Few-shot Pseudo-supervised Contrastive Learning, ICLR 2023 \
> [3] Lu et al., Self-supervision can be a good few-shot learner, ECCV 2022 \
> [4] Li et al., Unsupervised few-shot image classification by learning features into clustering space, ECCV 2022
>
> ---
>
> **[SG1 & 2]**: Suggestions on clarity about model parameter updating, and detailed comparative analysis with PsCo.
>
> **[ASG1 & 2]**: Thank you for the detailed suggestions. We have already addressed both thoroughly as discussed earlier, and reflections are inserted in the (updated) revised manuscript.

---

> ### Author Response · Authors · 2023-11-22
> **Last Remarks or Suggestions?**
>
> We are delighted to know that you are **very satisfied** with our efforts and responses. We thank you for all your constructive comments and greatly appreciate you increasing the score. Please note that the current version of the paper is **already updated twice** per your instructions, please kindly have a look. As you rightfully stated, we also do believe that _"$\texttt{BECLR}$ has the potential to redefine self-supervised learning and its implications for U-FSL, especially in few-shot scenarios, and holds substantial importance in advancing the field"_. In that light, we are determined to ensure that you are fully convinced about the impact of our work.
>
> Is there any last step we can take, any last remark or suggestion for us to further improve the draft accotdingly, so that you would support our paper more strongly? We are more than happy to do so. Please let us know.

---

### Official Review · Reviewer_msQP · 2023-10-31

**Soundness:** 3 good
**Presentation:** 3 good
**Contribution:** 3 good
**Rating:** 8
**Confidence:** 3

**Summary:**

This paper proposed an Unsupervised few-shot learning method (BECLR). The key idea is to extend the memory queue concept to dynamically updated memory clusters (DyCE). The second key idea is to address sample bias issue (distribution shift between the unlabeled Query set and the labeled support set) by introducing OpTA at inference time. BECLR is the name of the overall method.

**Strengths:**

The paper tried to address several issues within a single framework. It benefits from contrastive pre-training, tries to address and distribution shift and address some of the issues around the memory queue concept.
It seems that empirical results are strong.

**Weaknesses:**

-SAMPTransfer (Shirekar et al 2023) is also based on membership but its performance is reported only on the miniImageNet-->CDFSL task (Table 3). It is missing from other experiments.
-Prior FSL works that are related to the distribution shift (sample bias) issue are not discussed.
-Table 1: For ResNet-50 and Wide ResNet backbones, some of the comparison methods are missing. Again, in Table 3 some of the comparison methods are missing. It seems that the subset of methods used in each experiment is an arbitrary subset of the available pool of the previous methods.

**Questions:**

See above

---

> ### Author Response · Authors · 2023-11-16
> **Response to Reviewer msQP (1/3)**
>
> We thank the reviewer for their positive and constructive feedback. We are pleased to note that the reviewer finds our extensive experimentation results strong, which define $\texttt{BECLR}$ as the new SoTA across all existing U-FSL baselines. Following a thorough analysis of the reviewer's suggestions, we address the reviewers concerns and questions in a point-to-point fashion. The modifications applied to the (updated) revised draft are marked in $\textcolor{blue}{blue}$ for convenience.
>
> ---
> **[Q1]**: SAMPTransfer performance missing. \
> **[A1]**: Thanks for your insightful observation and remark.
> - First of all, please have a look at Tables $9$, $10$ and $11$, already existing in the Appendix C. **SAMPTransfer's performance on miniImageNet and tieredImageNet is reported in Table $9$ and its performance on miniImageNet → CDFSL in Table $11$** (extended versions of Tables $1$ and $3$ in the main text).
> - Regarding SAMPTransfer [1], following your suggestion, we setup the end-to-end pipeline to reproduce its results. We now know all its bells and whistles well. As we discussed already in Section 2 (Related Work to U-FSL) in the main text, SAMPTransfer's performance is hindered by its graph-based membership exploration unit and thus its performance can even degrade with larger backbones. We speculate that this is the reason why its performance is not reported on larger backbones in the original paper. To substantiate this and address your concern, we have conducted an additional experiment during this rebuttal period, comparing the performance of $\texttt{BECLR}$ and SAMPTransfer for different backbones. As seen in the table below: (i) SAMPTransfer indeed performs better with a shallower backbone (Conv4b) and (ii) $\texttt{BECLR}$ shows a superior end-to-end performance, regardless of the choice of backbone. Finally, **we have included the performance of SAMPTransfer on Table $1$ of the revised draft, following your remark**. \
> Would you also advise including the below Table in the Appendix of the revised draft?
>
> \begin{array}{lccccc}
> \hline
> && \text{miniImageNet}& \text{miniImageNet}\newline \hline
> \text{Method}& \text{Backbone}& \text{5 way 1 shot}& \text{5 way 5 shot}\newline \hline
> \texttt{SAMPTransfer}& \text{Conv4}& 55.75 ± 0.77& 68.33 ± 0.66\newline
> \texttt{SAMPTransfer}& \text{Conv4b}& 61.02 ± 1.05& 72.52 ± 0.68\newline
> \texttt{BECLR}& \text{Conv4b}& \textbf{69.54 ± 0.63}& \textbf{80.61 ± 0.29}\newline \hline
> \texttt{SAMPTransfer}& \text{ResNet-18}& 45.64 ± 0.73& 56.82 ± 0.65\newline
> \texttt{BECLR}& \text{ResNet-18}& \textbf{75.74 ± 0.62} & \textbf{84.93 ± 0.33}\newline \hline
> \end{array}
>
>
> [1] Shirekar et al., Self-Attention Message Passing for Contrastive Few-Shot Learning, WACV 2023

---

> ### Author Response · Authors · 2023-11-16
> **Response to Reviewer msQP (2/3)**
>
> **[Q2]**: Prior FSL works that are related to the distribution shift (sample bias) are not discussed. \
> **[A2]**: Great suggestion.
> - Let us first set the scene a bit better: as we articulate in the paper, the problem of sample bias (distribution shift between query and support sets) is _overlooked_ by almost every U-FSL baseline and the vast majority of the supervised FSL prior art. This is the very reason behind our Key Idea II in Section 1 (Introduction) and the proposition of the novel module called $\texttt{OpTA}$.
> - Following your suggestion, we have delved deeper in the FSL literature and found related work that attempts to alleviate the effects of sample bias. In particular, [1] proposes to enhance the support set by drawing additional base-class images via class-competitive attention maps, [2] highlights that sample bias is most severe when support embeddings are in the vicinity of the task centroid, whereas [3] utilizes a calibrated distribution for sampling more support embeddings, yet these methods explicitly rely on base-class characteristics. [4,5] follow a different perspective and tackle sample bias directly on the logistic classifier-level by using Firth bias reduction [6] techniques. These methods, however, are prone to overfitting of the classifier and pretrained encoder during fine-tuning. In contrast, our proposed $\texttt{OpTA}$ module requires no fine-tuning (and no additional trainable parameters) and does not depend on the pretraining dataset. Instead, we utilize optimal transport [7] for aligning the distributions of the _biased_ support set with that of the query set at the inference stage. **We have included this discussion in Section 2 (Related Work) of the revised version, along with further elaboration on the distribution shifts and biases that are present in the U-FSL setting.** We are happy to also include any specific references from the (U-)FSL literature you might find relevant and want us to refer to, please just advise and we will include.
>
> [1] Chen et al., Few-Shot Learning with Part Discovery and Augmentation from Unlabeled Images, IJCAI 2021 \
> [2] Xu et al., Alleviating the sample selection bias in few-shot learning by removing projection to the centroid, NeurIPS 2022 \
> [3] Yang et al., Free Lunch for Few-shot Learning: Distribution Calibration, ICLR 2021 \
> [4] Ghaffari et al., On the importance of firth bias reduction in few-shot classification, ICLR 2022 \
> [5] Wang et al., Revisit Finetuning strategy for Few-Shot Learning to Transfer the Emdeddings, ICLR 202 \
> [6] Firth, David, Bias reduction of maximum likelihood estimates. \
> [7] Cuturi et al., Computational optimal transport: With applications to data science.

---

> ### Author Response · Authors · 2023-11-16
> **Response to Reviewer msQP (3/3)**
>
> **[Q3-a]**: Some comparisons are missing in Table 1 and 3. It seems that the subset of methods used in each experiment is an arbitrary subset of the available pool of the previous methods.\
> **[A3-a]**: Thank you for this observation and remark. Let us further expand on this.
>
> - First of all, it is _virtually impossible_ for us to reproduce all existing baselines on all benchmarks. We did reproduce a good group of most recent ones, but for some we either could not even find an authentic repository or could not reproduce their reported results. As such, we naturally have to constrain ourselves to the performances and FSL benchmarks reported in the original paper, trusting the authors.
>
> - Secondly, FSL benchmarks have very specific, built-in data samplers allowing the baselines to precisely report their performances following those guidelines. For both our reproductions of existing baselines and $\texttt{BECLR}$'s reported results, **we have taken every measure to ensure fairness in our comparisons** by following the most commonly adopted pretraining and evaluation settings in the U-FSL literature in terms of pretraining/inference benchmark datasets (for both in-domain and cross-domain experiments), pretraining data augmentations, ($N$-way, $K$-shot) inference settings and number of query set images per tested episode.
>
> - We, nonetheless, report results of $\texttt{BECLR}$ on ALL existing FSL benchmarks out there, which is not the case for most baselines. **As such, if a certain baseline does not report performance on those benchmarks, and we could not find an authentic codebase nor reproduce their reported results, we had to exclude them from our reported results in the corresponding benchmark.** By the way, this is the case in most studies out there. Consequently, the reported subset of methods in each experiment is not arbitrary but rather dependent on these limitations.
>
> - Please also have a look at Tables $9$, $10$ and $11$, already existing in Appendix C. Therein, **we cover essentially ALL existing baselines on U-FSL** (to the best of our knowledge) as well as some from supervised FSL (just for bench-marking purposes). We are happy to also include additional results, in case you feel we have omitted any relevant U-FSL baseline, please feel free to share.
>
> ---
>
> **[Q3-b]**: For ResNet-50 and Wide ResNet backbones, some of the comparison methods are missing.\
> **[A3-b]**: Thank you for the remark and for observing this.
> - Given the inherent data scarcity in the U-FSL setting, most baselines report results on shallower backbones (e.g., Conv4, Conv5, ResNet-12, ResNet-18). Among these backbones, ResNet-18 has been most commonly adopted across the recent U-FSL literature [1-5], as such is the main backbone for our analysis, ensuring fairness in our comparisons. The reported results with a ResNet-50 backbone are provided in order to assess whether $\texttt{BECLR}$ can scale to deeper backbones. In this setting, we similarly had to limit the pool of of comparing methods to those, which report their performance with a ResNet-50 feature extractor.
>
> - The focus of our work is on U-FSL, and not supervised FSL. Vast majority of U-FSL approaches do not present results with Wide ResNet (WRN) backbones and some supervised FSL approaches do. The comparison against the selected supervised baselines is our best effort to present their _top performing_ models. Nevertheless, we understand this might create some inconsistency as you rightfully indicated. As such, **following your suggestion, we have (i) excluded supervised methods, trained with WRN, from our comparisons in Table 1 (ii) refactored Table 1, now comparing all baselines with identical backbone configuration (ResNet-18 and ResNet-50, when applicable) to ensure consistency and fairness in our comparisons.**
>
> [1] Lu et al., Self-supervision can be a good few-shot learner, ECCV 2022 \
> [2] Chen et al., Shot in the dark: Few-shot learning with no base-class labels, CVPR 2021 \
> [3] Wang et al., Contrastive prototypical network with Wasserstein confidence penalty, ECCV 2022 \
> [4] Li et al., Unsupervised few-shot image classification by learning features into clustering space,  ECCV 2022 \
> [5] Chen et al., Few-Shot Learning with Part Discovery and Augmentation from Unlabeled Images, IJCAI 2021

---

> > ### Author Response · Authors · 2023-11-20
> > **Gentle Follow up on Our Responses**
> >
> > We wanted to kindly make a follow up on the elaborate response and revisions we have provided and see if they address your concerns. We would be more than happy to provide further clarifications and revisions if you have any more questions or concerns, and if not we would greatly appreciate it if you would please re-evaluate our paper's score. Thank you again for your reviews which helped us tremendously to improve our paper!

---

> ### Author Response · Authors · 2023-11-22
> **Last Remarks or Suggestions?**
>
> Dear reviewer msQP,
>
> As the author-reviewer engagement window is closing out soon, and since we haven not heard back from you, we wanted to make a kind follow up to see if our elaborate responses and revisions address all your concerns. Notably, our tremendous effort and active engagement with the other two reviewers have already led to an increase in their overall score. **We are determined to ensure that you are also fully satisfied with our work and its impact**. As such, we would be more than happy to provide further clarifications and revisions if you have any more questions or concerns, and if not we would greatly appreciate it if you please re-evaluate our paper's final score.
>
> Thank you so much for all your constructive recommendations and insightful comments. \
> Authors

---

### Author Response · Authors · 2023-11-16
**General Response**

Dear Reviewers and AC,

We sincerely appreciate your invaluable time and effort invested in reviewing our manuscript. Your constructive and insightful feedback helped us improve the quality of the (updated) revised draft.

As reviewers highlighted, we propose a novel  approach (All Reviewers) with a well-explained and coherent narrative (Reviewers WH4q, wMvt) demonstrating state-of-the-art results across ALL existing FSL benchmark, substantiated through extensive experimentation (Reviewers WH4q, wMvt). As Reviewer WH4q highlights, $\texttt{BECLR}$ has the potential to redefine self-supervised learning and U-FSL and holds substantial importance in advancing the field.

In response to reviewer's comments, we have carefully revised and enhanced the manuscript with the following additional discussions and experimentation:

- We further elaborate on the limitations and differences of other baselines as well as on the unreported results where applicable, also add new results to partially cover that.

- After thoroughly investigating and reproducing PsCo's results, we now have additional discussions in the text and a new table comparing different aspects of $\texttt{BECLR}$ and PsCo.
- We have further elaborated on the importance and broader impact of the proposed distribution alignment module $\texttt{OpTA}$, with new results and explanations.

These updates are temporarily highlighted in "$\textcolor{blue}{blue}$" in the revised manuscript for your convenience to check. We hope our response and revisions sufficiently address all your concerns and suggestions.

Thank you very much.

Best regards, \
Authors.

---

### Comment · Area_Chair_TQaj · 2023-11-22
**Let's have more discussion with authors**

Dear reviewers,

The author-reviewer discussion period is closing at the end of Wednesday Nov 22nd (AOE). Let's take this remaining time to have more discussions with the authors on their responses to your reviews. Should you have any further opinions, comments or questions, please let the authors know asap and this will allow the authors to address them.

Kind regards,
AC

---

### Author Response · Authors · 2023-11-23
**Note to Reviewers**

Dear Reviewers,

Thanks for your constructive reviews. The rebuttal window will close out shortly, so we wanted to kindly follow up one last time to make sure your concerns are all properly addressed. **If all your concerns have been addressed and you are satisfied with our responses and efforts, we would greatly appreciate if you could please revisit and adjust your final rating and evaluation of our work.** Many thanks again for all your insightful comments.

Kind regards,\
Authors

---

### Comment · Area_Chair_TQaj · 2023-12-04
**Clarifying the requirement of OpTA**

Dear authors and reviewers,

Thank you for all the hard work to resolve issues related to this submission. Before we conduct Reviewer-AC discussion, I have a few additional questions related to the OpTA module.

1. To effectively implement OpTA, it seems this work assumes that $NQ$ query samples can be accessed at the same time (as a batch). Then the optimal transport plan $\pi^*$ can be inferred using Eq.(2). If it is this case, what if 1) the query samples arrive one by one or 2) there is only one or few query samples per class? Will the OpTA still work effectively in this situation?

2. Just want to confirm that when using Eq.(2) to infer the optimal transport plan $\pi^*$, this work does NOT accidentally access any information related to the true class labels of the query set (say, knowing which query samples are from the same class, etc.).

3. In Section 3, it is mentioned that "... contains $NQ$ (with $Q > K$) unlabeled samples." Why is the condition "$Q > K$" listed here? Is it related to the proposed OpTA?

Looking forward to hearing your response or comment.

Regards,
AC

---

### Meta-Review · Area_Chair_TQaj · 2023-12-08

**Metareview:**

Based on the submission, reviews, and author feedback, the main points that have been raised are summarised as follows.

Strengths:

1. It addresses several issues within a single framework, and the empirical results are strong and impressive.
2. It merges existing concepts innovatively and presents fresh modules.
3. The research is methodologically robust and the effectiveness is demonstrated by comparison.
4. It has the potential to redefine the implication of self-supervised learning for U-FSL and holds substantial importance.
5. The paper is detailed and easy to follow. Well motivated and substantiated by experiments and ablations.

Issues:

1. Prior FSL works that are related to the distribution shift (sample bias) issue are not discussed.
2. The proposed method bears a striking resemblance to PsCo. Shall provide a comprehensive comparison with it.
3. The evaluation should have considered the various components in separation and w.r.t. other methods.

The authors have done very well in providing detailed and informative feedback to address the above raised issues. Reviewers have read the feedback and responded to the authors positively. So far, all the ratings are leaning towards accepting this work. AC shares the similar opinion after reading the submission. This is a solid piece of work with new insights and contribution. Especially, the proposed OpTA scheme demonstrates consistent and impressive improvement. The authors shall maximally incorporate the answers in the author feedback into the submission, which will further strengthen this work.

**Justification For Why Not Higher Score:**

The contribution is not at the level of a new breakthrough or a new framework. Therefore, Oral is not recommended.

**Justification For Why Not Lower Score:**

1. This is a solid piece of work with new insights and contribution. For example, a reviewer commented that "It has the potential to redefine the implication of self-supervised learning for U-FSL and holds substantial importance."
2. Especially, the proposed OpTA scheme demonstrates consistent and impressive improvement. It could be interesting to many researchers working on self-supervised learning and few-shot learning. Spotlight will make them aware of this interesting piece of work.

Considering the above issues, spotlight is recommended.

---

### Decision · Program_Chairs · 2024-01-16

Accept (spotlight)